# Enforcing convex constraints in Graph Neural Networks

**Ahmed Rashwan** *
University of Bath

**Keith Briggs**
BT Research

**Chris Budd**
University of Bath

**Lisa Kreusser**
University of Bath and Monumo

## Abstract

Many machine learning applications require outputs that satisfy complex, dynamic constraints. This task is particularly challenging in Graph Neural Network models due to the variable output sizes of graph-structured data. In this paper, we introduce ProjNet, a Graph Neural Network framework which satisfies input-dependant constraints. ProjNet combines a sparse vector clipping method with the Component-Averaged Dykstra (CAD) algorithm, an iterative scheme for solving the best-approximation problem. We establish a convergence result for CAD and develop a GPU-accelerated implementation capable of handling large-scale inputs efficiently. To enable end-to-end training, we introduce a surrogate gradient for CAD that is both computationally efficient and better suited for optimization than the exact gradient. We validate ProjNet on four classes of constrained optimisation problems: linear programming, two classes of non-convex quadratic programs, and radio transmit power optimization, demonstrating its effectiveness across diverse problem settings.

## 1 Introduction

In many machine learning (ML) applications, model outputs must satisfy certain constraints to ensure feasibility, safety, or interpretability [22, 40]. A common approach to enforcing constraints is through activation functions (e.g., Sigmoid, Softmax) in the output layer [41], which restrict outputs to a predefined set. However, while effective for simple linear or box constraints, this approach is not applicable for more complicated types of constraints. Many real-world applications require complex, data-dependent constraints. For example, in robotics, constraints can encode feasible movements or collision avoidance [21, 54, 63]. Similarly, in energy systems, power generation limits vary with demand fluctuations [48], and in industrial processes, physical constraints must be respected [46]. Such tasks are often modelled as optimisation problems with convex constraints.

Several approaches have been proposed for enforcing a given set of convex constraints $C$ in ML models, each with distinct strengths and drawbacks. Many existing methods provide only approximate feasibility, typically by incorporating the constraints into the training loss using barrier functions or duality [22, 27, 33]. Instead, we will only be considering methods which yield strict feasibility guarantees. One common strategy involves first selecting an arbitrary reference point within $C$, and then using an ML model to compute an adjustment vector about this reference, ensuring that the final output remains feasible and accomplishes the desired task; we refer to this as the vector clipping method [39, 52, 59]. Typically, the reference point is obtained using a convex solver, which is generally more computationally expensive than evaluating the adjustment vector. While this approach works well for fixed constraint sets, it becomes significantly more costly when the constraints vary with the model input, as a new reference point must be computed for each instance. This limitation makes training vector clipping models with larger batch sizes impractical.

---

*email: ar3009@bath.ac.uk

39th Conference on Neural Information Processing Systems (NeurIPS 2025).

A related line of research explores the integration of differentiable optimisation layers within ML models [1, 3, 12]. These methods leverage convex solvers capable of solving batches of optimisation problems concurrently, with gradients computed analytically. As a result, optimisation layers are more efficient in handling input-dependent constraints and have applications beyond constraint satisfaction. However, a key limitation lies in their scalability to large problem instances. The solvers used typically rely on interior-point methods, which require matrix factorizations that are difficult to efficiently implement on GPUs, particularly for sparse matrices [4]. This makes it difficult to use this approach with GNNs, since graph-structured data is typically sparse and leads to heterogenous batches, requiring efficient exploitation of sparsity to sustain performance.

In addition, simple iterative algorithms have been embedded in ML models for constraint satisfaction, sharing similarities with our approach. A prominent example is the Sinkhorn algorithm [19], which has been used to compute stochastic matrices for learning permutations [24, 55] and for satisfying positive-linear constraints [61]. These iterative algorithms are well-suited for GPU implementations. However, their gradients are typically computed by unrolling the iterative procedure through back-propagation, which can be both computationally expensive and memory-intensive. We argue that unrolling is not always required: since many iterative algorithms—such as Sinkhorn—produce solutions to specific optimisation problems, their gradients can often be derived analytically, leading to more efficient methods of gradient computation [44].

Projection methods have long been used for finding feasible points in convex sets [16]. A classic example is Von Neumann's algorithm [6], which uses alternating projections to find a feasible point in the intersection of two convex sets. While this algorithm produces feasible points, it generally does not allow us to characterize *which* point in $C$ is returned. In contrast, Dykstra's projection algorithm [34] can find the unique projection of a point onto the intersection of convex sets. For our purposes, being able to determine the exact point produced by the algorithm is crucial, as it enables the computation of a surrogate gradient. The Component-Averaged Dykstra (CAD) algorithm [14] is a variant of Dykstra's method which exploits problem sparsity by using component averaged projections [15] on the constituent sets $C_i$. CAD can be efficiently accelerated using GPU scattering operations [35] and works seamlessly on batches of problems. We present a convergence result for the CAD and demonstrate its application to constraint satisfaction in ML models.

Building on these ideas, we introduce ProjNet, a GNN architecture designed for constraint satisfaction. ProjNet leverages the CAD algorithm to guarantee feasibility, while incorporating a sparse variant of the vector clipping method to improve model expressiveness. The architecture is end-end differentiable and fully GPU accelerated. Although this work focuses on linear constraints, most of the methods presented naturally extend to broader classes of convex constraints. We demonstrate the effectiveness of ProjNet across four classes of constrained optimization problems.

## 1.1 Contributions

Our key contributions are as follows:

1. We establish theoretical foundations of our work by proving a convergence result for the CAD algorithm, explicitly determining its limit, and incorporate these into an efficient GPU implementation, enabling significant speed-ups and scalability for large-scale applications.

2. We introduce a computationally efficiently surrogate gradient for the CAD algorithm, tailored for ML applications.

3. We present a refined variant of the vector clipping method, which we refer to as sparse vector clipping, which leverages problem sparsity and is well-suited for use in GNN models.

4. We propose ProjNet, a GNN architecture for constraint satisfaction problems which integrates the CAD algorithm with sparse vector clipping. Unlike many existing approaches, our method naturally supports batched graph inputs.

5. We demonstrate the effectiveness of our GNN-based approach on four constrained optimisation problems: linear programming, two classes of non-convex quadratic programs, and radio transmit power optimisation.

## 1.2 Overview

This paper is structured as follows. Section 2 describes the individual components of our GNN architecture. We study the CAD algorithm in Section 3, providing a convergence result, a surrogate gradient, and numerical experiments. In Section 4, the effectiveness of our method is demonstrated on four classes of optimisation problems. Finally, Section 5 concludes the paper.

## 2 ProjNet: A GNN framework for constraint satisfaction

We propose ProjNet, a GNN-based framework for solving constraint satisfaction problems. After introducing some preliminaries and notation in Section 2.1, we construct a constrained input graph $\mathcal{G}_C$ in Section 2.2 which embeds a set of linear output constraints $C$ into a given input graph $\mathcal{G}$. ProjNet is then defined as a model $\text{GNN}_\theta$ that maps any constrained input graph to a feasible solution, i.e., $\text{GNN}_\theta(\mathcal{G}_C) \in C$ for all $\mathcal{G}_C$. The architecture enforces constraint satisfaction through two core components: a projection layer and sparse vector clipping layers, detailed in Sections 2.3 and 2.4, respectively.

### 2.1 Preliminaries

**Convex constraint sets**   We define a convex, non-empty constraint set $C = \bigcap_{i=1}^{m} C_i \subset \mathbb{R}^n$ as the intersection of individual constraint, where each $C_i = \{x \in \mathbb{R}^n : g_i((x_j)_{j \in N_i}) \leqslant 0\}$. Here, $N_i \subset \{1, \ldots, n\}$ specifies the subset of variables affected by each constraint $C_i$, and $g_i : \mathbb{R}^{N_i} \to \mathbb{R}$ is a convex function. Given component $j \in \{1, \ldots, n\}$, we write $L_j = \{i : j \in N_i\} \subset \{1, \ldots, m\}$ for the set of constraints affecting $j$, and set $l_j = |L_j|$. We are especially interested in linear constraints where $g_i(x) = A_i x - b_i$ for $1 \leqslant i \leqslant m$, and $A_i$ denote the rows of some matrix $A = [A_1, \ldots, A_m] \in \mathbb{R}^{m \times n}$ and $b \in \mathbb{R}^m$. For linear constraints, the constraint set $C = \{x \in \mathbb{R}^n : Ax \leqslant b\}$ is a polytope. We denote the unique projection of a point $x$ onto a convex set $C$ by $P_C(x) = \arg\min_{z \in C} ||x - z||^2$.

**GNN inputs**   We assume that all problem instances are given as weighted graphs $\mathcal{G} = (X, E)$ where the rows of $X \in \mathbb{R}^{n \times k}$ correspond to $n$ nodes of $\mathcal{G}$, each with a $k$-dimensional node feature, and $E \in \mathbb{R}^{n \times n}$ encodes edge weights of all the edges connecting the $n$ nodes. The input of the GNN will be a constrained input graph $\mathcal{G}_C$ which will be associated with graph $\mathcal{G}$ and constraint set $C$, and will be defined in Section 2.2. We denote the output of the GNN by $y \in \mathbb{R}^n$ where each output component $y_i \in \mathbb{R}$ corresponds to the output feature of node $i$. Our aim is to enforce the constraint $y \in C$ where the constraint $C$ is input-dependant.

**Batched inputs**   GNNs can be simultaneously applied to a batch of constrained input graphs associated with a batch of input graphs $(\mathcal{G}_1, \ldots, \mathcal{G}_\beta)$ of the form $\mathcal{G}_i = (X_i, E_i)$ for $i \in \{1, \ldots, \beta\}$ by interpreting them as one large disconnected graph $\mathcal{G} = (\text{cat}(X_1, \ldots, X_\beta), \text{diag}(E_1, \ldots, E_\beta))$. Here, $\text{cat}$ denotes concatenation and $\text{diag}$ computes a block-diagonal matrix. This perspective allows GNNs and all other forms of graph computation discussed in this paper to generalise seamlessly to the batched case. Since these batched graphs are larger and more sparse than the individual input graphs $\mathcal{G}_i$, it is particularly important to develop methods that exploit their sparsity.

### 2.2 Constrained input graphs

Inspired by graph-based formulations in mixed-integer programming [17, 29], where a linear objective is combined with linear constraints into a single graph, we extend this methodology beyond linear objectives to more general tasks represented by some graph $\mathcal{G}$. Specifically, we construct a constrained input graph $\mathcal{G}_C$ from a given input graph $\mathcal{G}$ and a set of linear constraints $Ax \leqslant b$. This new graph $\mathcal{G}_C = (X, b, A, E)$ is heterogeneous, comprising variable nodes $X$, constraint nodes $b$, and two types of edges: $E$ and $A$. The graph contains $n$ variable nodes with $k$-dimensional features from the rows of $X$, and $m$ constraint nodes, each associated with a scalar feature $b_i$. Edges in $E$ represent relationships among variable nodes in $X$, while the matrix $A$ serves as a weighted adjacency matrix capturing connections between variable and constraint nodes. This unified graph representation enables the use of a single GNN to handle tasks that incorporate both problem instances and their associated constraints.

Some of the we methods we present exploit the sparsity of $\mathcal{G}_C$ by identifying independent groups of constraints. Specifically, we wish to identify the finest partition $\mathcal{P}$ of $\{1, \ldots, m\}$ such that constraints from different components of $P$ do not share any constrained variables. This partition corresponds to the connected components of the bipartite graph $(X, b, A)$ and can be computed efficiently using GPU graph processing software [62], or alternatively, via a simple label propagation scheme outlined in Appendix D.1.

## 2.3 Component-averaged Dykstra (CAD) algorithm

To ensure feasibility, we wish to compute the projection of any vector $x \in \mathbb{R}^n$ onto $C$, denoted by $P_C(x)$. To achieve this, we use the component-averaged Dykstra (CAD) algorithm [14], an iterative scheme which exploits problem sparsity and is well suited for GPU implementations. CAD belongs to a family of Dykstra-style projection algorithms, and is closely related to the two-set Dykstra algorithm [34] and the Simultaneous Dykstra Algorithm [38]. All of these methods operate under a common principle: computing the projection $P_C$ onto $C$ by using the projections $P_{C_i}$ on individual sets $C_i$. For $m = 2$, i.e. $C = C_1 \cap C_2$, the iterates of the two-set Dykstra algorithm are defined as:

$$
\begin{aligned}
y^{(k)} &= P_{C_1}(x^{(k)} + p^{(k)}), \\
p^{(k+1)} &= x^{(k)} + p^{(k)} - y^{(k)}, \\
x^{(k+1)} &= P_{C_2}(y^{(k)} + q^{(k)}), \\
q^{(k+1)} &= y^{(k)} + q^{(k)} - x^{(k+1)}.
\end{aligned} \tag{2.1}
$$

with initial values $x^{(1)} = x$ and $p^{(1)} = q^{(1)} = 0$. Assuming $C$ is non-empty, the sequence of points $x^{(k)}$ defined by (2.1) is guaranteed to converge to $P_C(x)$ [16, 34].

A variation of Dykstra's algorithm can be extended to $m > 2$ by using a product space formulation [38] to reduce the $m$ constraint setting to the two constraint case (2.1), referred to as the Simultaneous Dykstra Algorithm which holds for any $m \in \mathbb{N}$:

$$
\begin{aligned}
x^{(k+1)} &= \frac{1}{m} \sum_{i=1}^{m} P_{C_i}(x^{(k)} + p_i^{(k)}), \\
p_i^{(k+1)} &= x^{(k)} + p_i^{(k)} - P_{C_i}(x^{(k)} + p_i^{(k)}), \quad 1 \leqslant i \leqslant m,
\end{aligned} \tag{2.2}
$$

where $x^{(1)} = x$ and $p_1^{(1)} = \ldots = p_m^{(1)} = 0$. The Simultaneous Dykstra Algorithm is well-suited for GPU implementations as it only relies on simple vector operations and allows for parallel computation of the projections $P_{C_i}(x^{(k)} + p_i^{(k)})$. However, the convergence of (2.2) can be slow for sparse problems, as for a given constraint $C_i$, only a small number of components of $x^{(k)} + p_i^{(k)}$ and $P_{C_i}(x^{(k)} + p_i^{(k)})$ differ when $|N_i| \ll n$, and because of this, many components of the iterates $x_k$ only slowly change. To overcome the slow convergence of (2.2), instead of averaging over all constraints as in (2.2), averaging over constraints relevant for a given component has been proposed in [14, 15]. Then, for any $m \in \mathbb{N}$, the CAD algorithm yields

$$
\begin{aligned}
x_j^{(k+1)} &= \frac{1}{l_j} \sum_{i \in L_j} \left( P_{C_i}(x^{(k)} + p_i^{(k)}) \right)_j, \\
p_i^{(k+1)} &= x^{(k)} + p_i^{(k)} - P_{C_i}(x^{(k)} + p_i^{(k)}), \quad 1 \leqslant i \leqslant m.
\end{aligned} \tag{2.3}
$$

To the best of our knowledge, no convergence results are available for (2.3).

### 2.3.1 The linear CAD algorithm

Dykstra-type algorithms assume that individual set projections $P_{C_i}(x)$ can be directly computed. This is possible for many classes of constraint sets such as linear subspaces, half-spaces, and second-order cones [9]. Even when $P_{C_i}(x)$ is not known in closed-form, we can use approximate hyperplane projections [10]. For linear constraints of the form $C_i = \{x \in \mathbb{R}^n : A_i x \leqslant b_i\}$, these individual projections are given by

$$
P_{C_i}(x) = x + \min \left\{ 0, \frac{b_i - A_i x}{||A_i||^2} \right\} A_i^T. \tag{2.4}
$$

We refer to the CAD algorithm (2.3) with projections defined by (2.4) as the linear CAD algorithm [37]. One of the key advantages of linear CAD is that it naturally lends itself to GPU implementations as the sparse multiplications in (2.3) and (2.4) can be efficiently computed using GPU scattering operations [35]. This makes the linear CAD algorithm significantly more scalable than traditional optimisers for computing projections onto convex polytopes.

## 2.4 Sparse vector clipping

Given a feasible point $z \in C$, we propose a sparse vector clipping layer that transforms $z$ into a learned output $y \in C$ by leveraging the sparsity of $\mathcal{G}_C$. We start by learning an unconstrained direction vector $\text{GNN}_\theta^v(\mathcal{G}_C; z) = v \in \mathbb{R}^n$ first. For each constraint $C_i$, we want to determine the maximum scaling factor $\alpha_{C_i}$ so that $z + \alpha_{C_i} v \in C_i$, i.e. $\alpha_{C_i} = \max\{\alpha \geqslant 0 : z + \alpha_{C_i} v \in C_i\}$. Note that only nodes $j \in N_i$ are relevant for constraint $C_i$, and as such only $(z_j)_{j \in N_i}$ and $(v_j)_{j \in N_i}$ need to be considered in practice when finding $\alpha_{C_i}$. Computing each factor $\alpha_{C_i}$ is tractable for many types of convex sets $C_i$ and for many such scaling factors, closed-form equations are provided in [59].

The overall scaling factor is the minimum over the individual constraints $\alpha_C = \min\{\alpha_{C_1}, \ldots, \alpha_{C_m}\}$. The standard vector clipping scheme ensures feasibility by computing output $y = z + \min\{1, \alpha_C\}v \in C$, where $\min\{1, \alpha_C\}v$ is the clipped direction. As $C$ is convex, any point $y \in C$ can be reached from any feasible point $z \in C$ using the above construction. However, when $m$ is large, $\alpha_C$ can become very small, limiting how far the the output $y$ can move from $z$ and thus significantly reducing expressiveness.

To address this, we propose to exploit constraint independence. Let $\mathcal{P}$ be the partition of the constraints into independent sets, as defined in Section 2.2. For each component $p \in \mathcal{P}$, we compute a local scaling factor $\alpha_p = \min_{i \in p} \alpha_{C_i}$. This allows us to compute separate clipped directions for each independently constrained set of variables. More precisely, for each $p \in \mathcal{P}$, we compute output variables $y_j = z_j + \min\{1, \alpha_p\} v_j$ for all $j \in N_i$ where $i \in p$. . This definition is valid since, by construction, no two components of $\mathcal{P}$ constrain the same variable.

By applying scaling locally within each independent constraint group, this approach leverages the sparsity of $C$ to allow more expressive model outputs. The resulting vector $y \in C$ remains feasible, and the method is fully differentiable with respect to both the initial point $z$ and direction vector $v$.

To summarise, a sparse vector clipping layer $y = \text{SVC}_\theta(\mathcal{G}_C; z)$ procedes as follows:

1. Compute unconstrained direction $v = \text{GNN}_\theta^v(\mathcal{G}_C; z) \in \mathbb{R}^n$.

2. For each constraint $i$, compute individual factors $\alpha_{C_i}$, only depending on $(z_j)_{j \in N_i}$ and $(v_j)_{j \in N_i}$.

3. For each connected component $p \in \mathcal{P}$, compute factor $\alpha_p = \min_{i \in p} \alpha_{C_i}$.

4. Return output vector $y \in C$, where for each component $p \in \mathcal{P}$ and all constraints $i \in p$, the variables $j \in N_i$ are computed as $y_j = z_j + \min\{1, \alpha_p\} v_j$.

We can stack multiple sparse vector clipping layers to iteratively refine an input $z \in C$. For this, let $z^{(0)} = z \in C$ denote the input of the first layer and iteratively define $x^{(k+1)} = \text{SVC}_{\theta_k}(\mathcal{G}_C; x^{(k)})$, this leads to a set of $I + 1$ feasible points $z^{(0)}, \ldots, z^{(I)} \in C$. Notably, subsequent feasible points $(z^{(k)})_{k \geqslant 1}$ are significantly less computationally expensive than the initial projection $z^{(0)}$, which is obtained using the CAD algorithm.

## 2.5 Network architecture

A common technique for enforcing output constraints in machine learning involves computing an initial (unconstrained) output $w$ and then projecting it onto the feasible set $C$. As shown in [47], this projection-based strategy is a universal approximator for constrained functions. Our approach, called ProjNet, follows this principle by applying the CAD algorithm to $w$ which results in a feasible solution $z \in C$ to the constrained problem. Since $z$ lies on the boundary of $C$ when $w \notin C$, we propose to improve model expressiveness by making the interior of $C$ more accessible using sparse vector clipping layers, introduced in the previous section.

Given a constraint input graph $\mathcal{G}_C$ as input, ProjNet, visualised in Figure 1, consists of three steps:

1. GNN: Compute an unconstrained output $w = \mathrm{GNN}_\theta^w(\mathcal{G}_C) \in \mathbb{R}^n$ using a GNN over the constrained graph $\mathcal{G}_C$.

2. CAD Projection: Compute $z = P_C(w)$ using the GPU-accelerated CAD algorithm.

3. Sparse Vector Clipping: Compute a sequence of feasible points $z^{(0)}, \dots, z^{(I)} \in C$ using $I$ sparse vector clipping layers, starting from $z^{(0)} = z$ with model output $z^{(I)} = y$.

ProjNet is fully GPU-accelerated, enabling efficient processing of large-scale inputs $\mathcal{G}_C$. Both the CAD algorithm and sparse vector clipping layers are differentiable with respect to their input, making ProjNet fully end-to-end differentiable allowing us to learn $y$ using standard backpropagation. We will further show in Section 3.3 that it is possible to compute a surrogate gradient for projections without resorting to unrolling procedures.

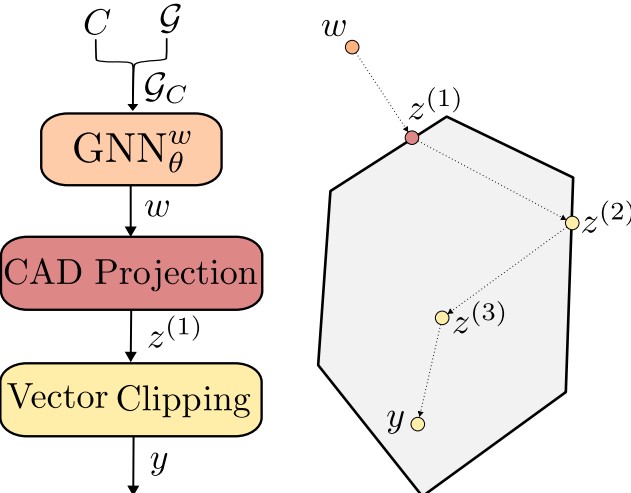

Figure 1: An illustration of the ProjNet architecture. Shows forward pass on a constraint polygon where points are colour coded with the modules used to compute them. Model takes as input a graph $\mathcal{G}$ with $n$ nodes and a set of linear constraints $C \subset \mathbb{R}^n$, and outputs a feasible point $y \in C$.

## 3 Analysis of the Component-averaged Dykstra (CAD) algorithm

### 3.1 Convergence of the CAD algorithm

While both the two-set (2.1) and simultaneous (2.2) Dykstra algorithms converge to $P_C(x)$, we show that the CAD algorithm (2.3) converges to a certain non-orthogonal projection determined by the sparsity structure of $C$. We show this by using a sparse product space formulation, demonstrating that (2.3) can be regarded as a special case of (2.1) under a certain transformation.

**Theorem 1.** *For input $x \in \mathbb{R}^n$, the CAD algorithm (2.3) converges to the projection $P_C^l(x) = \arg\min_{y \in C} \sum_{j=1}^n l_j(y_j - x_j)^2$. In particular, for input point $(x_j/\sqrt{l_j})_{1 \leq j \leq n}$ and feasible set $\{(x_j/\sqrt{l_j})_{1 \leq j \leq n} : x \in C\}$, the CAD algorithm converges to $((P_C(x))_j/\sqrt{l_j})_{1 \leq j \leq n}$.*

We prove Theorem 1 in Appendix A.1. To use the CAD algorithm for computing the orthogonal projection $P_C(x)$, we scale by its inputs by $1/\sqrt{l_j}$ and rescale the outputs by $\sqrt{l_j}$. We incorporate this in our GPU implementation of CAD outlined in Appendix D.2.

### 3.2 Computational efficiency

We compare the linear CAD algorithm's runtime against Gurobi [31] – a popular commercial solver – for computing the projection $P_C(x)$ on polytopes $C$. We used the dual barrier method for Gurobi as we found it to be the fastest amongst the available options for the class of problems considered. Our results in Figure 2 demonstrate that the CAD algorithm can solve projection problems up to two orders of magnitude faster than Gurobi when the both the numerical tolerance is relaxed and

the initial point $x$ is close to $C$. However, this advantage diminishes significantly when either of these conditions is not met. To encourage the unconstrained vector $w$ in ProjNet to be closer to the feasible set, we add an additional penalty term $c_h \|w - P_C(w)\|^2$ to the loss function during training. The hyperparameter $c_h$ provides a trade-off between speed and performance: increasing $c_h$ speeds up the CAD algorithm by moving $w$ closer to $C$, but may reduced task performance.

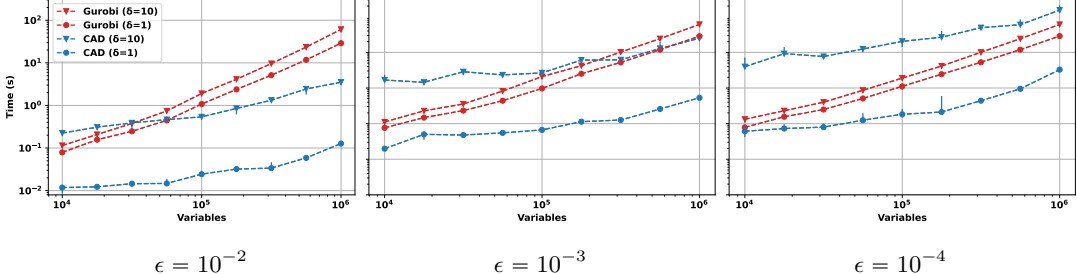

$$\epsilon = 10^{-2} \qquad\qquad \epsilon = 10^{-3} \qquad\qquad \epsilon = 10^{-4}$$

Figure 2: Comparing runtimes of CAD algorithm and Gurobi for the linear projection problem as a function of output dimension $n$. $\delta$ values in the legend are a measure of the distance between the initial point and the feasible set. Legend and y-axis are identical for all figures.

### 3.3 Surrogate gradients

To embed the CAD algorithm into ML models, its derivative is required during the backward pass which —according to Theorem 1— is identical to the derivative of $P_C$. In this subsection, we first present the exact gradient of $P_C$ before introducing a surrogate gradient that is both computationally cheaper and more effective for optimization compared to the exact gradient. Surrogate gradients have proven useful in scenarios where the exact gradient either doesn't exist or is unsuitable for optimisation, particularly in spiking neural networks [32, 49] and straight-through estimators [7].

For fixed linear constraints $C$, let $H_x \subset \mathbb{R}^n$ be the set of all points orthogonal to $x - P_C(x)$. Define the tangent cone [53] at $P_C(x)$, denoted by $T_x \subset \mathbb{R}^n$, as the set of all directions $v \in \mathbb{R}^n$ that can be approached by moving toward points in $C$ from $x$. More formally, $v$ is in $T_x$ if there exist sequences $(x_i)_{i \in \mathbb{N}} \subset C$ and $(\tau_i)_{i \in \mathbb{N}} \subset \mathbb{R}_+$ such that $x_i \to x$, $\tau_i \to 0$ and $\frac{x_i - x}{\tau_i} \to v$.

Since $P_C$ is the projection onto a polytope, it is piecewise-affine and therefore differentiable almost everywhere. Its Jacobian $\partial_x P_C$ is given by the linear projection $P_{T_x \cap H_x}$ [26] which can be computed using the CAD algorithm – for instance. However, this gradient is not appropriate for training ML models, as it can be expensive to compute and often has low rank, which may lead to poor local optima during optimization.

To remedy, we propose to use a surrogate gradient by projecting onto $H_x$ only, namely

$$\partial_x P_C \approx P_{H_x} = \begin{cases} I - d_x\, d_x^T, & x \notin C, \\ I, & x \in C, \end{cases} \tag{3.1}$$

where $d_x = (x - P_C(x)) / \|x - P_C(x)\|_2$ and $I$ denotes the identity operator.

Proposition 1 states some desirable properties of this surrogate gradient, proved in Appendix A.2:

**Proposition 1.** *Let $C \subset \mathbb{R}^n$ be a polytope defined by non-redundant linear inequalities and let $x \in \mathbb{R}^n$ be a point where $P_C$ is differentiable. Then the surrogate Jacobian $P_{H_x}$, defined in (3.1), satisfies the following properties:*

1. ***Rank guarantee:*** $\operatorname{rank}(P_{H_x}) \geq n - 1$, *while no such bounds exist for* $\operatorname{rank}(\partial_x P_C)$.

2. ***Exactness:*** $P_{H_x} = \partial_x P_C \iff x \in \operatorname{int}(C)$, *or $P_C(x) \in \operatorname{int}(C_i)$ for all but one half-space $C_i$.*

3. ***Alignment with descent direction:*** *For any $v \in \mathbb{R}^n$, $\langle \partial_x P_C(v), P_{H_x}(v) \rangle = \|\partial_x P_C(v)\|^2$, indicating that $P_{H_x}(v)$ aligns with the descent direction of the true projected gradient.*

4. ***Local equivalence of gradient steps:*** *There exists $\beta > 0$ such that $P_C(x + \partial_x P_C(v)) = P_C(x + P_{H_x}(v))$ for all $v \in \mathbb{R}^n$ with $\|v\| < \beta$. This indicates that gradient-descent steps are identical for surrogate and exact gradients for a sufficiently small step-size.*

Table 1: Average objective values and their standard deviation for each application problem. For linear programming, optimality is measured instead. Last row combines programming methods. Data is obtained for different training seeds and problem instances.

| Type | Method | Linear | Quadratic (ER) | Quadratic (BA) | Transmit power |
|------|--------|--------|----------------|----------------|----------------|
| ML | ProjNet ($c_h = 0$) | $0.9973 \pm 0.0001$ | $6.83 \pm 1.47$ | $7.73 \pm 1.52$ | $\mathbf{3.01 \pm 0.23}$ |
| | ProjNet ($c_h = 0.01$) | $0.9905 \pm 0.0004$ | $6.78 \pm 1.31$ | $7.86 \pm 1.59$ | $2.82 \pm 0.36$ |
| | ProjNet ($c_h = 1$) | $0.9632 \pm 0.0015$ | $6.75 \pm 1.50$ | $7.67 \pm 1.38$ | $1.79 \pm 0.31$ |
| | No SVC ($c_h = 0$) | N/A | $6.76 \pm 1.67$ | $7.72 \pm 1.47$ | $2.92 \pm 0.26$ |
| | No CAD | $0.2954 \pm 0.7644$ | $1.93 \pm 1.33$ | $1.91 \pm 1.16$ | $1.79 \pm 0.30$ |
| | No SVC or CAD | $0.3365 \pm 0.2662$ | $1.69 \pm 1.29$ | $1.43 \pm 1.04$ | $1.71 \pm 0.30$ |
| Classic | trust-constr | N/A | $\mathbf{6.86 \pm 1.51}$ | $\mathbf{7.98 \pm 1.68}$ | $2.15 \pm 0.59$ |
| | LP / DCP / FP | $\mathbf{1.0000 \pm 0.0000}$ | $6.38 \pm 1.80$ | $7.60 \pm 1.70$ | $2.79 \pm 0.61$ |
| | Random | $0.1585 \pm 3.6216$ | $0.05 \pm 1.07$ | $0.00 \pm 0.80$ | $1.59 \pm 0.27$ |

Numerical experiments in Appendix E demonstrate that the surrogate gradient is both faster and more stable than the exact gradient when $C$ is a polytope. While beyond the scope of this work, we conjecture that the benefits of the surrogate gradient (3.1) extend to general convex constraints $C$.

## 4 Numerical results

To assess the performance of the ProjNet model, we test it on four classes of linearly constrained optimisation problems. For comparison, we employ both classical optimization-based methods, which are detailed for each specific task, as well as ML baselines in the form of ablations of the two main components of ProjNet: sparse vector clipping (SVC) and CAD. For reference, we also included a method which samples a random point in $C$ using the hit-and-run algorithm [43]. All methods considered are guaranteed to produce feasible outputs $y \in C$.

The problem instances used in our evaluation were randomly generated using different graph distributions, as outlined in Appendix B. In Figures 3 and 4, we report the runtime for selected methods as a function of problem size.

Average objective values are shown in Table 1. For linear programming, we present the ratio of achieved-to-optimal objective values, while for the other three applications, we report average objective values. Consequently, the standard deviations for linear programming appear smaller, since they are normalized by each instance's optimal value obtained from an LP solver.

### 4.1 Linear programming

Linear programs (LPs) are perhaps the simplest non-trivial class of convex problems. They are a well-studied class of problems with very broad applications and some highly optimised solvers [28].

Most algorithms for LPs focus on obtaining numerically precise, exact solutions. Such methods are usually CPU-based as they utilise sparse linear algebra computations that are not easily accelerated using GPUs. Being reliant on CPU computation makes such classical methods less scalable to large LPs, for this reason there has been some recent interest in using first-order methods to solve LPs [5, 64], but state-of-the-art solvers remain mostly rooted in CPU computation. We consider the problem of finding fast, approximate solutions to large-scale LPs of the form

$$\max_x \ c^T x, \quad \text{s.t. } Ax \leqslant b. \tag{4.1}$$

**Classical baselines** We consider three LP solvers: Gurobi, a widely used commercial solver; PDLP [5], a recent first-order method for large-scale LPs; and PDLP-GPU, its GPU-accelerated implementation. We use the gurobipy interface for Gurobi, OR-Tools [51] for PDLP, and CVXPY [20] for PDLP-GPU, which interfaces with NVIDIA's cuOpt solver .

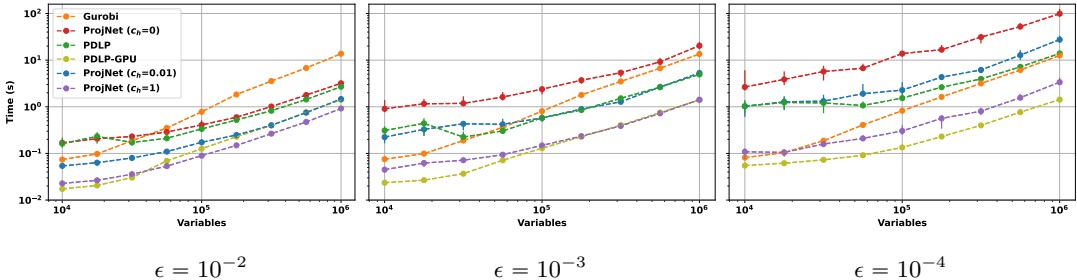

$$\epsilon = 10^{-2} \qquad\qquad \epsilon = 10^{-3} \qquad\qquad \epsilon = 10^{-4}$$

Figure 3: Comparing runtime of ProjNet, PDLP, and Gurobi for linear programming. We trained three ProjNet models with different $c_h$ values shown in legend. Error bars show upper/lower quartiles for each point. Legend and y-axis are identical for all figures.

## 4.2 Non-convex quadratic programming

We consider quadratic programs of the form

$$\max_x \ x^T Q x + c^T x, \quad \text{s.t. } Ax \leqslant b. \tag{4.2}$$

If $Q$ is negative-definite then the problem is convex and can be solved efficiently using interior-point solvers [9]. Alternatively, Dykstra-type algorithms can also be directly used to solve convex quadratic problems [38], although this is not a popular approach due to its slower convergence. We will instead be considering the general, non-convex case in which $Q$ is an arbitrary symmetric matrix.

Quadratic problems of the form (4.2) can be modelled as weighted graphs $\mathcal{G} = (c, Q)$, where the linear coefficients $c$ are treated as node features and the matrix $Q$ defines undirected edge weights. To evaluate our approach, we tested two types of graph topologies for $\mathcal{G}$: Erdős-Rényi (ER) [25] and Barabási-Albert (BA) [2] random graphs.

**Classical baselines** We tested a Difference of convex programming (DCP) method, which yields approximate solutions to (4.2) by solving a sequence of approximating convex quadratic programs [42], and trust-constr, a trust region method which yields feasible approximate solutions [13] implemented in the scipy.minimize package [60].

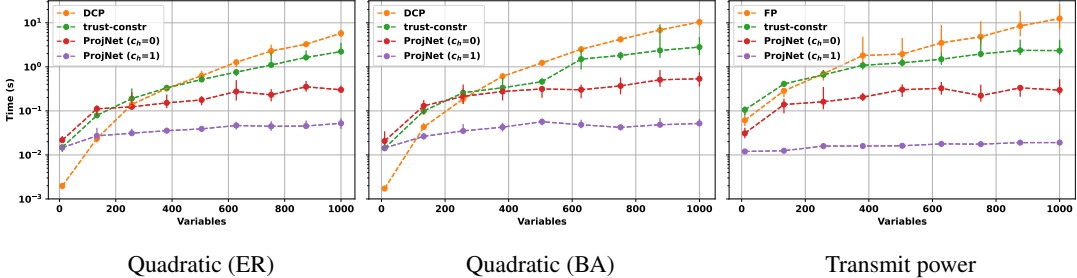

Quadratic (ER)      Quadratic (BA)      Transmit power

Figure 4: Comparing runtime of ProjNet, trust-constr, and programming baselines for three classes of optimisation problems. We show two ProjNet models with different values for $c_h$. Precision is set to $\epsilon = 10^{-3}$. Plots include error bars showing upper/lower quartiles for each point. Y-axis is shared.

## 4.3 Transmit power optimisation

Given $n$ radio transmitters, let $H \in \mathbb{R}^{n \times n}_{\geqslant 0}$ be a channel gain matrix and let $\sigma > 0$ denote background noise. We study the problem of optimising a vector of radio transmit powers $0 \leqslant x_i \leqslant p_{\max}$ such that the sum of channel capacities

$$c_i = \log\left(1 + \frac{H_{ii} x_i}{\sum_{j \neq i} H_{ij} x_j + \sigma^2}\right)$$

is maximised, subject to individual minimum capacity requirements $s_i > 0$:

$$\max_{x \in \mathbb{R}^n} \ \frac{1}{n} \sum_i c_i, \quad \text{s.t. } s_i \leqslant c_i, \ 0 \leqslant x_i \leqslant p_m. \tag{4.3}$$

This is a linearly constrained problem with a non-convex objective which can be shown to be NP-hard in general [18]. We focus on feasible approximate solutions to (4.3).

**Classical baselines**   We tested trust-constr [13], and Fractional programming (FP) [57], which uses a quadratic transform that yields a sequence of convex optimisation problems whose solutions converge to a local minimum to (4.3)

### 4.4 Discussion

As a result of its GPU-acceleration, ProjNet demonstrates a clear advantage in runtime over classical optimization methods on large-scale problems. When it comes to solution quality, ProjNet is very competitive with classical methods while outperforming them for transmit power optimization.

As anticipated in Section 3.2, the hyperparameter $c_h$ governs the trade-off between accuracy and speed, enabling users to tailor ProjNet's performance to specific application requirements. Ablation baselines confirm that both the SVC and CAD components offer a meaningful increase in performance over standard alternatives, with CAD providing the most substantial performance gains.

## 5   Conclusion

We presented ProjNet, a GNN architecture designed to solve constraint satisfaction problems involving large-scale, input-dependent linear constraints. ProjNet leverages the CAD algorithm, for which we present a convergence result, a GPU implementation, and an efficient surrogate gradient for training ML models that embed the CAD algorithm.

We evaluated the ProjNet architecture on four classes of linearly constrained optimisation problems. In comparison to both classical and ML baselines, we demonstrate that ProjNet offers a strong and tunable trade-off between computational efficiency and solution quality.

## 6   Acknowledgments

We gratefully acknowledge the support of the EPSRC Programme Grant EP/V026259/1, 'The mathematics of Deep Learning'.

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

# A  Proofs

## A.1  Convergence proof of the CAD algorithm

*Proof of Theorem 1.* Let $x^{(k)}, p_i^{(k)}$ denote the CAD iterates with initial point $x \in \mathbb{R}^n$ and $m$ convex constraints $C_i \subset \mathbb{R}^n$. We show that there exists two convex sets $\mathbf{C}_1, \mathbf{C}_2 \subset \mathbb{R}^{m \times n}$ and an initial point $\mathbf{X} \in R^{m \times n}$ which, when used as inputs for the two-set Dykstra algorithm 2.1, produce iterates $\mathbf{X}^{(k)}, \mathbf{P}^{(k)}, \mathbf{Q}^{(k)} \in \mathbb{R}^{m \times n}$ such that for all $k$ we have

$$x^{(k)} = T(\mathbf{X}^{(k)}),$$
$$p_i^{(k)} = \mathbf{P}_i^{(k)}, \quad 1 \leqslant i \leqslant m \tag{A.1}$$

where $T$ is a certain linear bijection. We do this by using a sparse product space formulation to describe the component-averaging process as a linear projection. Using this idea, we can describe both the simultaneous projections and component averages computed in (2.3) as projections onto convex sets. The convergence result for CAD iterates $x^{(k)}$ then follows from the convergence of $\mathrm{x}^{(k)}$ to $P_{\mathbf{C}_1 \cap \mathbf{C}_2}(\mathbf{X})$.

Let $\mathbf{C}_1 = \bigtimes_{i=1}^m C_i \subset \mathbb{R}^{m \times n}$ be the Cartesian product of sets $C_i$, formulated as a set of matrices. For any $c = (c_1, \ldots, c_m) \in \mathbf{C}_1$, we interpret row $c_i \in C_i$ as variable values satisfying constraint $C_i$. Hence, $\mathbf{C}_1$ encodes the problem constraints, and is the set of all possible variable values satisfying the constraints. Let $\mathbf{X}_i \in \mathbb{R}^n$ denote the $i$-th row of matrix $\mathbf{X} \in \mathbb{R}^{m \times n}$. Then, the projection of $X$ onto $\mathbf{C}_1$ is simply the projection of its rows onto the indivisual constraint sets:

$$P_{\mathbf{C}_1}(\mathbf{X}) = [P_{C_1}(X_1), \ldots, P_{C_m}(X_m)] = \bigtimes_{i=1}^m P_{C_i}(\mathbf{X}_i),$$

where we use the cross symbol to denote the stacking of row vectors into a matrix.

Define the set $\mathbf{C}_2 \subset \mathbb{R}^{m \times n}$ as the collection of matrices $\mathbf{X}$ such that:

- $\mathbf{X}_{ij} = 0$ if variable $j \notin N_i$, i.e., $j$ is not involved in constraint $C_i$,

- $\mathbf{X}_{ij} = \mathbf{X}_{kj}$ for all $i, k$ such that $j \in N_i \cap N_k$, i.e., a shared variable $j$ has consistent values across all constraints they appear in.

In other words, $\mathbf{C}_2$ encodes the sparsity and variable-sharing structure of the constraints $C$: only relevant variables appear in each row, and any variable shared across constraints must take the same value in those rows.

We now show that the linear projection $P_{\mathbf{C}_2}(\mathbf{X})$ is the component average over the columns of $\mathbf{X} \in \mathbb{R}^{m \times n}$. For this, we define basis matrices $\mathbf{V}^{(k)} \in \mathbb{R}^{m \times n}$ for each variable $k \in \{1, \ldots, n\}$, where each $\mathbf{V}^{(k)}$ encodes which constraints involve variable $k$. Specifically, the entries of $\mathbf{V}^{(k)}$ are given by:

$$\mathbf{V}_{ij}^{(k)} = \begin{cases} 1, & \text{if } j = k \text{ and } k \in N_i, \\ 0, & \text{otherwise.} \end{cases}$$

That is, the $i$-th row of $\mathbf{V}^{(k)}$ is nonzero only if constraint $C_i$ depends on variable $k$, and the nonzero entry appears in column $k$. The subspace $\mathbf{C}_2 \subset \mathbb{R}^{m \times n}$ has an orthonormal basis given by the scaled matrices $(l_k^{-1/2} \mathbf{V}^{(k)})_{k=1}^n$, where $l_k$ denotes the number of constraints involving variable $k$. The linear projection is then given as

$$P_{\mathbf{C}_2}(\mathbf{X}) = \sum_{j=1}^n \frac{\mathbf{V}^{(j)}}{l_j} \left\langle \mathbf{V}^{(j)}, \mathbf{X} \right\rangle$$
$$= \sum_{j=1}^n \frac{\mathbf{V}^{(j)}}{l_j} \sum_{i \in L_j} \mathbf{X}_{ij}.$$

There is a natural correspondence between elements $\mathbf{X} = \sum_{j=1}^{n} \gamma_j \mathbf{V}^{(j)} \in \mathbf{C}_2$ and vectors $x = [\gamma_1, \ldots, \gamma_n]^T \in \mathbb{R}^n$, defined by the linear bijection $T \colon \mathbf{C}_2 \to \mathbb{R}^n$,

$$T\left(\sum_{j=1}^{n} \gamma_j \mathbf{V}^{(j)}\right) = [\gamma_1, \ldots, \gamma_n]^T.$$

This map identifies the representation of a matrix in $\mathbf{C}_2$ with its coefficient vector in $\mathbb{R}^n$.

For an initial input $x \in \mathbb{R}^n$, let $\mathbf{X} = T^{-1}(x) \in \mathbf{C}_2$. The projection of $\mathbf{X}$ onto the constraint sets followed by projection back onto $\mathbf{C}_2$, and mapped to $\mathbb{R}^n$ by $T$, yields:

$$T \circ P_{\mathbf{C}_2} \circ P_{\mathbf{C}_1}(\mathbf{X}) = T \circ P_{\mathbf{C}_2}\left(\bigtimes_{i=1}^{m} P_{C_i}(\mathbf{X}_i)\right) = T\left(\sum_{j=1}^{n} \frac{\mathbf{V}^{(j)}}{l_j}\left\langle \mathbf{V}^{(j)}, \bigtimes_{i=1}^{m} P_{C_i}(\mathbf{X}_i)\right\rangle\right)$$

The inner product $\left\langle \mathbf{V}^{(j)}, \bigtimes_{i=1}^{m} P_{C_i}(\mathbf{X}_i)\right\rangle$ simplifies to:

$$\left\langle \mathbf{V}^{(j)}, \bigtimes_{i=1}^{m} P_{C_i}(\mathbf{X}_i)\right\rangle = \sum_{i \in L_j}(P_{C_i}(\mathbf{X}_i))_j = \sum_{i \in L_j}(P_{C_i}(x))_j,$$

Hence,

$$T \circ P_{\mathbf{C}_2} \circ P_{\mathbf{C}_1}(\mathbf{X}) = T\left(\sum_{j=1}^{n} \frac{\mathbf{V}^{(j)}}{l_j}\sum_{i \in L_j}(P_{C_i}(x))_j\right) = \left(\frac{1}{l_j}\sum_{i \in L_j}(P_{C_i}(x))_j\right)_{j=1}^{n}. \qquad \text{(A.2)}$$

Note that the right-hand side of (A.2) coincides with the component-average in (2.3). We can now show two equivalences (A.1). For $k = 1$ both equivalences hold by definition. For the induction step, we assume they hold at iteration $k-1$ and show they also hold at iteration $k$. For the first equivalence we have:

$$x^{(k)} = T \circ P_{\mathbf{C}_2} \circ P_{\mathbf{C}_1}(\mathbf{X}^{(k-1)} + \mathbf{P}^{(k-1)})$$
$$= T \circ P_{\mathbf{C}_2}\left(P_{\mathbf{C}_1}(\mathbf{X}^{(k-1)} + \mathbf{P}^{(k-1)}) + \mathbf{Q}^{(k-1)}\right)$$
$$= T(\mathbf{X}^{(k)})$$

where the second equality follows from the fact that $\mathbf{Q}^{(k-1)}$ is orthogonal to $\mathbf{C}_2$. To show the second equivalence, recall that for all $i \in \{1, \ldots, m\}$:

$$p_i^{(k)} = x^{(k-1)} + p_i^{(k-1)} - P_{C_i}(x^{(k-1)} + p_i^{(k-1)}).$$

For all $j \notin L_i$, we have $p_{ij}^{(k)} = 0$ and since $x_j^{(k-1)} = \mathbf{X}_{ij}^{(k-1)}$ for all $j \in L_i$ we have

$$p_i^{(k)} = \left(\mathbf{X}^{(k-1)} + \mathbf{P}^{(k-1)} - P_{\mathbf{C}_1}(\mathbf{X}^{(k-1)} + \mathbf{P}^{(k-1)})\right)_i = \mathbf{P}_i^{(k)}.$$

The convergence of (2.1) implies the convergence of the CAD iterates $x^{(k)}$ to the projection

$$T \circ P_{\mathbf{C}_1 \cap \mathbf{C}_2}(\mathbf{X}) = T\left(\underset{\mathbf{Y} \in \mathbf{C}_1 \cap \mathbf{C}_2}{\arg\min} \|\mathbf{Y} - \mathbf{X}\|^2\right)$$
$$= T\left(\underset{\mathbf{Y} \in \mathbf{C}_1 \cap \mathbf{C}_2}{\arg\min} \sum_{j=1}^{n}\sum_{i \in L_j}(\mathbf{Y}_{ij} - \mathbf{X}_{ij})^2\right)$$
$$= \underset{y \in C}{\arg\min} \sum_{j=1}^{n} l_j(y_j - x_j)^2 = P_C^l(x).$$

Finally, we derive the limit of the CAD algorithm for input $(x_j/\sqrt{l_j})_{1 \leqslant j \leqslant n}$. By scaling $x$ and $C$, we can use algorithm (2.3) to compute the orthogonal projection $P_C(x)$ instead. Given any two vectors $x, y \in \mathbb{R}^n$ with $y_i \neq 0$ and a set $C \subset \mathbb{R}^n$, let $x/y$ denote element-wise division and let $C/y = \{x/y : x \in C\}$. Setting $l = (l_1, \ldots, l_n)$ and using inputs $x/\sqrt{l}$, $C/\sqrt{l}$ in (2.3) converges to

$$P_{C/\sqrt{l}}^l\left(x/\sqrt{l}\right) = \underset{y \in C/\sqrt{l}}{\arg\min} \sum_i (\sqrt{l_i}\, y_i - x_i)^2 = \left(\underset{y \in C}{\arg\min} \|x - y\|^2\right)/\sqrt{l} = P_C(x)/\sqrt{l}.$$

Therefore, we obtain the projection $P_C(x)$ using the CAD algorithm (2.3) by scaling $x$ and $C$ by $1/\sqrt{l}$, and scaling the output by $\sqrt{l}$. $\qquad \square$

## A.2   Properties of the surrogate Jacobian

*Proof.* We prove each part of the proposition separately.

1. **Rank guarantee.** As $x - P_C(x) = 0$ for $x \in C$, this yields $\mathrm{rank}(P_{H_x}) = n$. For $x \notin C$, we have $x - P_C(x) \neq 0$, implying that the direction vector $d_x = \frac{x - P_C(x)}{\|x - P_C(x)\|}$ is well-defined. Then $P_{H_x} = I - d_x d_x^T$ is the orthogonal projection onto the hyperplane normal to $d_x$. Since $d_x d_x^T$ is a rank-1 matrix, $P_{H_x}$ has rank $n - 1$.

   It can be seen that there are no lower bounds on the rank of the exact gradient $\partial_x P_C$. We can demonstrate this using a simple counter-example. Let $C \subset \mathbb{R}^n$ be the unit hyper-cube defined by the inequality $x \leqslant \mathbb{1}$. For $x = 2 \cdot \mathbb{1}$ we have $P_C(x) = \mathbb{1}$ which is a vertex of $C$. The tangent cone and supporting hyperplane at $x$ are given by

$$T_x = \{x : x \leqslant 0\}$$
$$H_x = \{x : \sum_i x_i = 0\}.$$

   These sets intersect only at the origin, therefore $\mathrm{rank}(\partial_x P_C) = 0$.

2. **Exactness.**

   We prove each direction of the equivalence.

   - **Case 1:** $x \in \mathrm{int}(C)$. Then $P_C(x) = x$, so $d_x = 0$ and the surrogate projection becomes $P_{H_x} = I$. The projection map is locally the identity, so the Jacobian $\partial_x P_C = I$ as well. Hence, $P_{H_x} = \partial_x P_C$.
   - **Case 2:** $P_C(x) \in \mathrm{int}(C_i)$ for all but one half-space $C_i$. In this case, $P_C(x)$ lies on the boundary of exactly one of the half-spaces defining $C$, and in the interior of the rest. Then, locally around $x$, the constraint set behaves like a single active hyperplane. Therefore, the projection $P_C$ is locally the orthogonal projection onto that hyperplane. Since the Jacobian $\partial_x P_C$ in this case is the projection matrix onto the corresponding hyperplane, and $H_x$ is defined as the hyperplane orthogonal to $x - P_C(x)$, we again have $P_{H_x} = \partial_x P_C$.
   - **Converse:** If $P_{H_x} = \partial_x P_C = P_{T_x \cap H_x}$, it must be that $H_x \subset T_x$. This occurs when no constraints are active (i.e., $x \in \mathrm{int}(C)$), since then $H_x = T_x$. If $x \notin C$ then then $H_x$ is a hyperplane , which could only be contained in $T_x$ if the boundary of $C$ at $P_C(x)$ is locally of dimension $n - 1$. Clearly, this only occurs if only one constraint set is active, i.e. when $P_C(x) \in \mathrm{int}(C_i)$ for all but one $i$.

3. **Alignment with descent direction.** Let $v \in \mathbb{R}^n$. Note that $\partial_x P_C(v) \in T_x \cap H_x$, where $T_x$ is the tangent cone at $P_C(x)$, and $H_x$ is the hyperplane orthogonal to $x - P_C(x)$. Since $P_{H_x}$ is the orthogonal projection onto $H_x$, and $\partial_x P_C(v) \in H_x$, applying the projection preserves the vector:

$$P_{H_x}(\partial_x P_C(v)) = \partial_x P_C(v).$$

   Therefore, we have:

$$\langle \partial_x P_C(v), P_{H_x}(v) \rangle = \langle \partial_x P_C(v), \partial_x P_C(v) \rangle = \|\partial_x P_C(v)\|^2 .$$

4. **Local equivalence of gradient steps:** Since $P_C$ is piece-wise affine and differentiable at $x$, there exists some $\beta > 0$ such that $P_C$ is affine in the $\beta$-neighbourhood of $x$. In this affine region we have

$$P_C(x + y) = P_C(x) + P_{T_x \cap H_x}(y).$$

   for any $y \in \mathbb{R}^n$ with $\|y\| < \beta$. Moreover, since projections are non-expansive we have

$$\|\partial_x P_C(y)\| \leqslant \|P_{H_x}(y)\| \leqslant \|y\| < \beta.$$

   Therefore, both surrogate and exact gradient steps are in the affine region of $P_C$. We then compute

$$
\begin{aligned}
P_C(x + P_{H_x}(y)) &= P_C(x) + P_{T_x \cap H_x} \circ P_{H_x}(y) \\
&= P_C(x) + P_{T_x \cap H_x}(y) \\
&= P_C(x) + P_{T_x \cap H_x} \circ P_{T_x \cap H_x}(y) \\
&= P_C(x + \partial_x P_C(y)).
\end{aligned}
$$

$\square$

## B  Generating test problems

In this section, we describe the distributions of test problems which were used to train and test each of the three application problems described in Section 4.

### B.1  Generating problem objectives

Components of the linear objective terms in (4.1) and (4.2) were sampled i.i.d uniformly as $c_i \sim U(-1, 1)$. The sparse quadratic objective term in (4.2) was generated by first determining its sparsity pattern, choosing each element $Q_{ij}$ as non-zero with probability $d/n$. Each non-zero element was sampled uniformly as $Q \sim U(-10, 10)$. Finally, for the transmit power task, the channel gain matrix was generated by sampling a geometric graph to determine the sparsity pattern of $H$ then setting non-zero entries to $H_{ij} = (d_{ij} + 1)^{-3}$ where $d_{ij}$ is the distance between nodes $i$ and $j$ in the geometric graph. This is a standard distribution to use for transmit power problems [22].

### B.2  Generating linear constraints

We used sets of sparse, randomly generated linear constraints of the form $Ax \leqslant b$ which we now describe. Let $d$ denote the average degree of the adjacency matrix $A \in \mathbb{R}^{m \times n}$. To determine the sparsity pattern of $A$, we use the method described in the pervious section with degree $d - 1$, and then setting one additional random non-zero element $A_{ij}$ in each row $i$ to ensure no constraints are redundant . The value of $d$ determines the average degree of the corresponding bi-partite constraint graph. Then, each column $A_i$ is sampled as random unit length vector satisfying the chosen sparsity pattern. Finally, $b = u + As$ where $u_i \sim U(0.1, 1)$ and $s_i \sim \mathcal{N}(0, 1)$.

Our random linear constraints produce non-empty bounded polytopes such that a ball of radius 0.1 is guaranteed to fit inside $C$. Consequently, although the test problem distribution is appears fairly diverse, all instances are well-scaled and feasible. We leave the handling of poor scaling and infeasibility to future work.

## C  Stopping conditions

Stopping conditions are an essential part of any optimisation algorithm as they ensure that the computed solution is within a specified tolerance to the true solution while minimising the number of required iterations. We discuss stopping conditions for the CAD algorithm and outline some issues with comparing solvers which use different stopping conditions.

### C.1  CAD conditions

As shown in [8], Dykstra's algorithm can be stopped by using a certain monotone sequence. However, we decided to use the feasibility condition $\max\{Ax^{(i)} - b\} \leqslant \epsilon$ for a given precision $\epsilon > 0$ which appears to work well empirically. We provide a brief discussion of this condition and leave a more detailed theoretical analysis for future work.

For an initial point $x$ and CAD iterates $x^{(i)}$, we observed numerically that the sequence $(||x - x^{(i)}||_2)_i$ is strictly increasing. We are not aware of this result in the literature but, assuming it does hold, we may use the feasibility stopping condition $\max\{Ax^{(i)} - b\} \leqslant \epsilon$ since the condition is then satisfied with $\epsilon = 0 \iff x_i = P_C(x)$ as

$$||x - P_C(x)||_2 < ||x - y||_2, \quad \forall y \in C \backslash \{P_C(x)\}.$$

Moreover, Hoffman's lemma [36] guarantees the existence of an error bound of the form

$$d(x_i, C) = \min_{y \in C} ||x^{(i)} - y||_2 \leqslant c \max\{Ax^{(i)} - b\}$$

for some constant $c > 0$ which depends only on $C$. This allows us to place lower and upper bounds on the dual solution $x - P_C(x)$ as

$$||x - x^{(i)}||_2 \leqslant ||x - P_C(x)||_2 \leqslant ||x - x^{(i)}||_2 + d(x^{(i)}, C)$$
$$\leqslant ||x - x^{(i)}||_2 + c \max\{Ax^{(i)} - b\}.$$

### C.2   Comparing solvers

When comparing different optimisation algorithms at a given numerical precision, it is often difficult to equate numerical tolerance parameters $\epsilon$ due to differences in stopping conditions. We only use a primal feasibility condition for the CAD algorithm while most solvers use both primal and dual feasibility conditions. Gurobi appears to only use absolute values of $\epsilon$ while PDLP uses both absolute and relative $\epsilon$'s. The comparisons we offered are thus not completely precise but we would expect the general trends to hold.

## D   Implementation details

In this section, we discuss some practical details regarding our numerical experiments. All experiments were conducted on a single machine with an NVIDIA GeForce RTX 4090 GPU and an AMD Ryzen 9 7950X3D CPU.

### D.1   Computing graph connected components

For a given bipartite graph with $n$ nodes of one type and $m$ nodes of another, we describe a simple label propagation method for computing the set $P$ of connected components. Initializing node labels as $l^{(1)} = (1, \ldots, m)$ and iteratively updated them as

$$l_i^{(k+1)} = \min_{j \in N_i} l_j^{(k)}.$$

Each label $l_i^k$ is the smallest node index in the $k$-hop neighbourhood of node $i$, therefore for any two connected nodes $i, j \in \{1, \ldots, n\}$ there exists some $k' > 0$ such that $l_{k,i} = l_{k,j}$ for all $k \geqslant k'$. This sequence of labels converges to a vector $l \in \mathbb{R}^m$ where all constraints in the same connected component share the same label. This algorithm can be efficiently implemented using GPU scattering operations, described below.

### D.2   CAD implementation

Our GPU implementation of the CAD algorithm relies on the **scatter** function which is used to compute inhomogeneous sums. Efficient implementations of **scatter** are provided by most popular ML frameworks including Pytorch [50] and TensorFlow [45]. Given an index array $I \in \mathbb{N}^n$ and a value array $V \in \mathbb{R}^n$, scattering computes

$$(\mathbf{scatter}(I, V))_i = \sum_{j : I_j = i} V_j.$$

Scattering allows us to compute both the half-space projections (2.4) and component inhomogeneous averages without directly resorting to sparse matrix methods, which we found to be slower.

The full linear CAD algorithm is shown in Algorithm 1. The algorithm takes a sparse representation of $A$ as input where $A_{\text{row}}, A_{\text{col}}$ denote the row and column indices of the non-zero values of $A$ with values $A_V$, and includes tolerance $\epsilon$. The matrix $A$ is assumed to contain at least one non-zero element in each row, as otherwise that row is redundant. The while loop in lines 9-15 includes the GPU-accelerated CAD iterates in (2.3) while lines 3, 5, and 16 perform a change of variables to ensure that the algorithm converges to the projection $P_C(x)$ as opposed to the non-orthogonal projection, based on Theorem 1. Algorithm (1) is executed concurrently for each independent set of constraints $p \in \mathcal{P}$ with independent stopping conditions.

**Algorithm 1** The linear CAD algorithm

---

1: **Input:** $A_\text{row}, A_\text{col}, A_V, b, x, \epsilon$
2: $l \leftarrow \textbf{scatter}(A_\text{col}, \mathbb{1})$
3: $A_V \leftarrow A_V \cdot \sqrt{l}[A_\text{row}]$
4: $A_\text{norm} \leftarrow \sqrt{\textbf{scatter}(A_\text{row}, A_V^2)}$
5: $x \leftarrow x/\sqrt{l}$
6: $A_V \leftarrow A_V/A_\text{norm}$
7: $b \leftarrow b/A_\text{norm}$
8: $p \leftarrow 0$
9: **while** $\max\{Ax - b\} > \epsilon$ **do**
10: $\quad x_V \leftarrow x[A_\text{col}]$
11: $\quad z \leftarrow x_V + p$
12: $\quad s \leftarrow \min\{b - \textbf{scatter}(A_\text{row}, A_V \cdot z), 0\}$
13: $\quad p \leftarrow -A_V \cdot s[A_\text{row}]$
14: $\quad x \leftarrow \textbf{scatter}(A_\text{col}, z - p)\,/\,l$
15: **end while**
16: $x \leftarrow x \cdot \sqrt{l}$
17: **Output:** $x$

---

### D.3 Computational efficiency experiments

The results in Figure 2 were obtained for randomly generated initial points $x$ where components are sampled as $x_i \sim U(-\delta, \delta)$, and random linear constraints $Ax \leqslant b$. Random constraints where sampled as described in B, but without the offset $u$. This ensures that the the origin-centred ball with radius 0.1 is feasible. The parameter $\delta$ is hence a measure of the average distance of $x$ to $C$. Results were gathered for different values of $\delta$ and the tolerance parameter $\epsilon$. Five data points where sampled for each triple $(\delta, \epsilon, n)$ tested. Besides Gurobi, we also tested the Clarabel [30] and OSPQ [58] solvers but found them to be slower for this class of problems.

### D.4 GNN architecture

GNN message passing layers on constrained graphs $G_C$ were computed using three separate message passes between the two different node types in $G_C$. Using the notation $X \rightarrow Y$ to denote a bipartite graph convolution which passes features of nodes $X$ to nodes $Y$, we computed the following message-passes in order:

1. Variables $\rightarrow$ Constraints

2. Variables $\rightarrow$ Variables

3. Constraints $\rightarrow$ Variables

Each of these message-passes was performed using a Graph Attention Network with dynamic attention [11].

For linear programming, we trained ProjNet models with 8 message passing layers for $\text{GNN}_\theta^v$ and no SVC layers. We didn't use any SVC layers since LPs always have solutions on the boundary of their feasible set, hence the boundary bias of projections doesn't negatively impact performance. As a result, the no SVC baseline wasn't relevant for LPs. For all other application problems we used 8 message passing layers and 3 SVC layers. As GNN and SVC components are fairly shallow, the CAD algorithm was the most expensive element of our ProjNet models Architectures of ablation baselines were adjusted to ensure that all models had a roughly equal number of learned parameters.

### D.5 Ablation experiments

We provide more details regarding the three ablation models considered. These models were obtained by replacing the CAD projection and sparse vector clipping (SVC) component in ProjNet. As a substitute for SVC, we used a standard form of the vector clipping scheme which instead directly uses the full constraint scaling value $\alpha_C$ and as a substitute for the CAD algorithm we instead took

$z$ to be a point in the interior $C$ which was computed by solving the following linear program

$$z = \arg\max_{x \in C}(\min\{b - Ax\}).$$

Runtime results for these ablation models were omitted from the main paper as they were generally not competitive to ProjNet in terms of their accuracy-speed trade-off. We provide more complete runtime results in Figure 5.

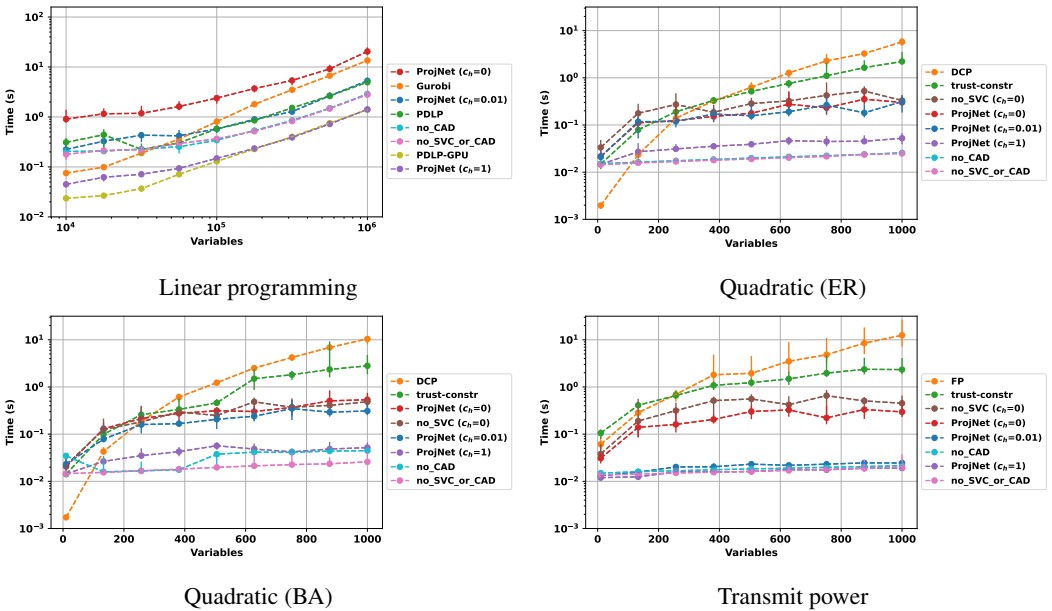

Figure 5: Plots showing method runtime as a function of number of number of variables $n$ for all baseline methods and application problems considered. Error bars denote upper/lower quartiles. Numerical tolerance was set to $\epsilon = 10^{-3}$.

### D.6 Difference of convex programming

The DCP method for solving non-convex quadratics is based on a heuristic method outlined in [42]. For a problem of the form (4.2), we write $Q = D - C$ where $D$ is a diagonal matrix with entries $D_{ii} = \sum_j c \cdot |Q_{ij}|$ where $c > 1$. Choosing $D$ this way ensures that $C$ is diagonally dominant, and hence $-C$ is negative-definite. Let $x_0 \in \mathbb{R}^n$, by linearising the diagonal matrix $D$, we can obtain a heuristic solution to (4.2) by solving the following sequence of convex problems:

$$x^{(n+1)} \in \arg\max_x \{x^T(c + Dx^{(n)}) - x^T C x \ : \ Ax \leqslant b\}. \tag{D.1}$$

These sub-problems were solved using the PIQP solver [56].

## E   Surrogate gradients experiments

In Figure 6, we provide empirical results comparing the exact and surrogate gradients from Section 3.3 for training ProjNet models. The exact gradient was computed using using the CAD algorithm with subspace constraint sets $C_i$, which in this case reduces to the component-averaging method presented in [23].

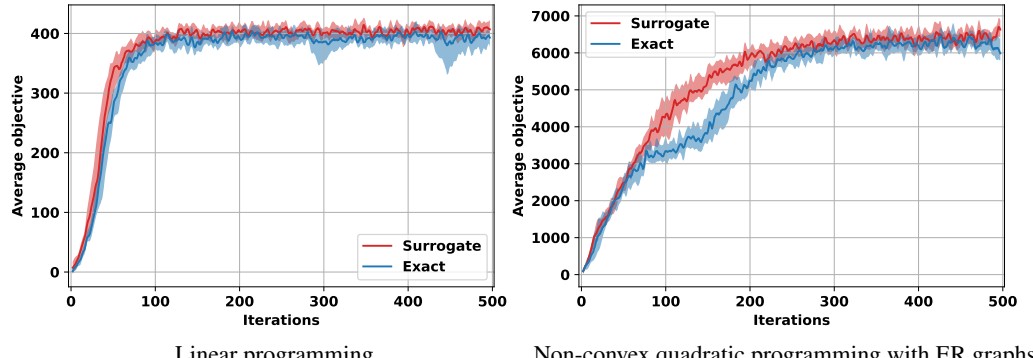

| Linear programming | Non-convex quadratic programming with ER graphs |

Figure 6: Training curves comparing exact and surrogate gradients to train ProjNet models for solving linear and non-convex quadratic programs. Error regions denote upper/lower quartiles. Curves are smoothed for visibility.

