# OpenReview forum: "Enforcing convex constraints in Graph Neural Networks"
_NeurIPS.cc/2025/Conference — NeurIPS 2025 poster_

### Official Review · Reviewer_RXdy · 2025-06-02

**Clarity:** 2
**Significance:** 3
**Originality:** 3
**Rating:** 4
**Confidence:** 3

**Summary:**

This work proposed a projection and search framework, using CAD and SVC. The architecture first ouputs an unconstrained output, then projects it to feasible region, and performs a search in the feaible region. The authors provided convergence analysis, designed a surrogate gradient for the CAD and exhibited promising experiment results.

**Questions:**

- Is it a supervised learning loss to predict the objective value?
- How is the SVC part trained? I mean how is the loss designed?
- For LPs, the numbers in the table are ratio of reaching optimal value? But for continuous optimization, how can you reach optimality? There must be some numerical error.
- What is the runtime bottleneck of the ProjNet? The CAD or MPNN or what?
- Each CAD step calculation seems cheap, yes, but if it takes a lot of steps then GPU cannot accelerate it.

**Ethical Concerns:**

["NO or VERY MINOR ethics concerns only"]

**Final Justification:**

I think this paper is novel, well written and exhibits good results. The main concern is the bottleneck of CAD component.
After consideration on the pros and cons, as well as other reviews, I personally believe borderline accept is a fair score.

**Limitations:**

The authors discussed the limitation in the paper.
The frameworks applies to linear constraints, which is a limitation.

**Quality:**

3

**Strengths And Weaknesses:**

Strength:
- The paper is well motivated. Even readers with little background knowledge can understand the research direction and target.
- The paper is well written.
- The project then search idea is interesting and clear, and seems easily implementable in practice.
- The practical consideration for CAD convergence and SVC step size is nice and very convincing.
- The experiment results are promising.

Weakness:
- Some implementation details are unclear, see questions below
- There is a concurrent work https://arxiv.org/abs/2502.02446, which also uses projection and search method for feasible solutions. Maybe consider citing it and add comparisons or discussions to their method.

---

> ### Author Rebuttal · Authors · 2025-07-29
>
> We thank the reviewer for their thoughtful and encouraging feedback. We are glad that you found the paper well-written, clearly motivated, and the projection-and-search framework both interesting and practical. We also appreciate the specific questions and suggestions regarding implementation details, comparisons to concurrent work, and limitations. We address each point in detail below and have incorporated clarifications into the revised manuscript.
>
> $\textbf{Weaknesses}$
>
> $\textbf{Implementation details:}$ We address your questions regarding implementation details in the Question section below.
>
> $\textbf{Related work:}$ We thank the reviewer for bringing this concurrent work to our attention. We have reviewed the cited paper [arXiv:2502.02446] and agree that it shares some similarities with our No CAD baseline (Appendix D.5). Their approach uses an interior-point (IP) solver to obtain a feasible initial point $x_0$, which is then refined using a GNN while maintaining feasibility—an idea aligned with projection-and-search strategies.
> While a direct empirical comparison is not currently available, we note that ProjNet is likely to be faster. As shown in our experiments, CAD achieves significantly lower runtime than IP solvers, and our method avoids the need for a full interior-point solve at inference time. Furthermore, unlike their approach, ProjNet is trained end-to-end, and includes a dedicated surrogate gradient to improve the backward pass.
>
> We will cite and discuss this work in the revised manuscript.
>
> $\textbf{Questions}$
>
> $\textbf{Learning loss:}$ Our method is trained in an unsupervised manner by directly maximizing the objective function (plus a penalty term $c_h ||w - P_C(w)||^2$), without requiring ground-truth labels. For instance, in the linear programming task, we train ProjNet to maximize the known objective $c \cdot x$, based solely on problem data.
>
> $\textbf{SVC training:}$ We do not design a separate or explicit loss for the SVC component. The entire ProjNet architecture, including the SVC layer, is end-to-end differentiable and trained jointly using a single overall loss function (e.g., the objective function combined with a penalty term for constraint proximity). As a result, the SVC component learns naturally during training, in the same way layers in a standard neural network (e.g., MLP) are optimized via backpropagation.
>
> $\textbf{LP numerical error:}$ All optimality ratios reported are measured relative to the solution obtained by PDLP, which we treat as a reference point. In our experiments, PDLP was run with both absolute and relative optimality tolerances set to various values of $\epsilon$, as detailed in the paper. While this does introduce a small numerical error, the resulting solutions are sufficiently accurate to serve as a practical proxy for the ground truth. Moreover, the optimality gaps observed for other methods are typically much larger than this numerical margin, so the comparisons remain valid and informative.
>
> $\textbf{Bottleneck:}$ The runtime bottleneck depends on several factors, including the size of the GNN, the number of SVC layers, and the value of the penalty weight $c_h$. In our setup, we used a relatively small GNN and at most 3 SVC layers as described as in Appendix D.4. Under these conditions, the CAD projection step was the primary bottleneck. For example, in the linear programming experiments with $c_h = 1$, the GNN was approximately 3$\times$ faster than CAD, making CAD the dominant cost component.
>
> $\textbf{GPU acceleration:}$ The reviewer is correct that the GPU does not reduce the number of CAD iterations. The GPU implementation only accelerates each iteration. To reduce the total number of steps, we use the penalty term $c_h ||w - P_C(w)||^2$ during training to encourage the initial point $w$ to lie closer to the feasible set $C$, thereby speeding up convergence. For instance, at $c_h=0, n=10^{4}, \epsilon=10^{-3}$ CAD needed $\sim$15k iterations, at $c_k=1$ that number dropped to about 300.
>
> $\textbf{Limitations}$
>
> While our experiments focus on linear constraints, our method extends to a broader class of convex constraints. Specifically, Theorem 1 is stated for arbitrary convex constraint sets, and the vector clipping technique we use can be (and has been, e.g., in [54]) applied to more general settings. The primary challenge lies in representing such constraints in a way that is amenable to GNNs or ML models more generally. We will clarify in the revised manuscript that our framework is not limited to linear constraints and, in principle, applies to arbitrary convex constraint sets.

---

> > ### Comment · Reviewer_RXdy · 2025-08-02
> >
> > Thank you for the reply, they indeed address my concerns.
> > However, the number of iterations of CAD and the fact that it is the bottleneck are the biggest limitations, in my opinion. It would restrict the applicability of this work.

---

> > > ### Author Response · Authors · 2025-08-03
> > >
> > > We appreciate the reviewer’s follow-up. We agree that the number of CAD iterations can be a limiting factor in some scenarios, and we will clarify this further in the revision. However, the iteration count is highly tunable via the penalty parameter $c_h$, and in our experiments, adjusting $c_h$ reduced the number of CAD steps by over 90% with often minimal impact on performance. Moreover, each CAD iteration is GPU-accelerated and typically takes less than 1 ms, resulting in a total runtime that remains practical even for large-scale problems.
> > >
> > > While CAD may be the primary bottleneck in our current setup (particularly with small models), this is not unique to our approach. Other techniques for constraint satisfaction, such as classical projection schemes or differentiable optimization layers, also incur significant computational cost, as noted in [3]. From this perspective, ProjNet offers a competitive and scalable alternative.

---

> ### Comment · Reviewer_RXdy · 2025-08-09
>
> Thank you so much for the clarification. I am curious how the parameter of CAD iterations would affect the precision. But considering the time limit, I would look forward to your final version. Besides, the projection method to ensure feasibility is not totally novel idea. I still believe the CAD part limits the applicability of this paper.
> As I said, I believe this paper definitely has its value, yet not conclusively should be accepted. I believe borderline accept would be a reasonable decision.

---

### Official Review · Reviewer_ev9W · 2025-06-22

**Clarity:** 3
**Significance:** 2
**Originality:** 3
**Rating:** 4
**Confidence:** 4

**Summary:**

This paper introduces ProjNet, a GNN architecture designed to enforce input-dependent constraints. To implement this architecture, Sparse Vector Clipping and Component-Averaged Dykstra are proposed for refined solution qualities. A surrogate gradient for CAD enables end-to-end training without unrolling. The authors provide a theoretical convergence guarantee for CAD and implement it with GPU acceleration. ProjNet is evaluated on several synthetic datasets involving both convex and non-convex problems

**Questions:**

In addition to the points raised in the weaknesses section, another concern is the omission of a potentially relevant baseline. The overall structure of ProjNet is to predict an initial solution, followed by post-processing to improve feasibility and optimality, which bears resemblance to [1] that follows a similar approach for integer programs.
While [1] focuses on discrete optimization, its method can be adapted to linear problems by relaxing the integrality constraints. A brief empirical or conceptual comparison would strengthen the paper. Moreover, this raises an important question about the individual contributions of the CAD and SVC components: could the predicted solutions from ProjNet serve as warm starts for established solvers like PDLP, potentially achieving comparable performance with less architectural complexity?

[1] Qingyu Han, et. al., A gnn-guided predict-and-search framework for mixed-integer linear programming. arXiv preprint arXiv:2302.05636, 2023.

**Ethical Concerns:**

["NO or VERY MINOR ethics concerns only"]

**Final Justification:**

I appreciate the author’s thoughtful and thorough response. All of my concerns have been adequately addressed, and I will maintain my original score.

**Limitations:**

yes

**Quality:**

3

**Strengths And Weaknesses:**

**Strengths**
1. The paper provides a convergence guarantee for the CAD algorithm and introduces a well-motivated surrogate gradient for training.
2. The GPU-accelerated implementation and sparse methods enable ProjNet to handle large-scale, input-dependent constrained problems efficiently.
3. ProjNet achieves surprisingly good results on non-convex optimization tasks, showing its practical utility.

**Weaknesses**
1. The main results are established on randomly generated instances, which is as convincing as using public datasets, especially considering.
2. The performance can vary significantly by adjusting the hyperparameter $c_h$, and not much details are provided.
3. While the paper claims a GPU-accelerated implementation of the CAD algorithm, it does not explicitly report whether the experimental results are based on the GPU version. If so, comparing against CPU-only baselines like PDLP may be unfair, as PDLP also supports GPU acceleration. Clarifying this would improve the fairness and transparency of the comparisons.

---

> ### Author Rebuttal · Authors · 2025-07-29
>
> We thank the reviewer for their thoughtful feedback. We are glad you found the surrogate gradient, convergence guarantee, and GPU implementation of CAD to be valuable contributions, and that the performance on non-convex tasks was promising. We address the questions and concerns about baselines, dataset design, hyperparameters, and GPU usage in detail below, and we have clarified these aspects in the revised manuscript.
>
> $\textbf{Weaknesses:}$
>
> $\textbf{Public datasets:}$ While we agree that using public datasets would be ideal, to the best of our knowledge, existing publicly available optimization datasets are typically small and likely not suitable for training general-purpose ML models. To support the development of scalable neural solvers, we generated a broad and diverse set of feasible problem instances tailored to our setting. This allowed us to control key properties (e.g., sparsity, constraint structure) while ensuring sufficient scale for training. Details of the generation process are provided in Appendix B.
>
> $\textbf{Penalty hyperparameter:}$ The hyperparameter $c_h$ controls the strength of the penalty term $c_h \|w - P_C(w)\|^2$ and encourages the GNN output $w$ to lie close to the constraint set $C$, thereby accelerating convergence of the CAD projection (see Figure 6).
>
> We tested values $c_h \in \\{0, 0.01, 0.1, 1\\}$ and observed a consistent trend: larger values of $c_h$ led to faster CAD convergence, but could interfere with task performance if set too high. Specifically, $c_h = 0.1$ reduced runtime compared to $c_h = 0.01$, but resulted in slightly worse solution quality. For our main experiments, we found $c_h = 0.01$ to offer a good trade-off between CAD speed and task accuracy. In practice, we recommend this as a robust default. The choice of $c_h$ will depend on the objective function's scale and can be adjusted depending on the desired trade-off between efficiency and performance.
>
> $\textbf{GPU-acceleration:}$ We thank the reviewer for raising this important point. We confirm that the experimental results reported in the paper use our GPU-accelerated implementation of the CAD algorithm.
>
> Regarding fairness, we note that most classical solvers we compare against do not have accessible or mature GPU implementations. For example, while PDLP supports GPU acceleration, we were unable to find a public implementation with a Python interface we can use. The OR-Tools package only offered a CPU implementation of PDLP. Nvidia is developing a GPU implementation, but it was still in beta and not publicly available at the time of our experiments.
>
> Moreover, traditional methods like the simplex or interior-point algorithms are challenging to parallelize efficiently due to complex control flow and sparse matrix operations, which limits practical GPU acceleration -- though this is an active area of research. In contrast, our method was specifically designed to benefit from GPU acceleration, which is central to its performance gains.
>
> To ensure transparency, we will clarify in the revised version that CAD was run on GPU, and we acknowledge this difference in platform when interpreting comparisons with CPU-only baselines.
>
> $\textbf{Questions:}$
>
> We thank the reviewer for highlighting [1] as a related approach. While [1] focuses on discrete optimization (specifically, MILPs), its structure of GNN-guided prediction followed by solution refinement does share high-level similarities with our approach. However, there are important distinctions. [1] does not offer true feasibility guarantees and still relies on classical solvers for completing the MILP solution. In contrast, our method ensures feasibility through the CAD projection and is much more general. We view our work as offering a stronger guarantee of constraint satisfaction, which is essential in many real-world applications. For linear programming, SOTA methods include simplex, interior-point, and (more recently) PDLP methods, all of which were tested. We are not aware of any competitive ML algorithms.
>
> Regarding warm-starting: These methods can work only if established solver exist for the problem considered. Meanwhile, ProjNet can be directly trained using any objective. Also, most established solvers do not support GPU acceleration, so even with a warm-start, they would likely be slow for large-scale problems -- as observed in the classical solver results we reported.
>
> We will add a short discussion of [1] in the revised related work section to acknowledge this connection.

---

> ### Comment · Reviewer_ev9W · 2025-08-05
>
> Thank you for making clarifications to the raised concerns, but I still have the following questions:
> 1. I agree that existing datasets are still insufficient for training truly general-purpose models. However, the coefficient ranges generated by the current experimental setup appear to be quite limited. Suppose the ultimate goal is to develop a general-purpose neural solver. In that case, it should be capable of handling a wide variety of problems, including those with highly diverse numerical ranges, as seen in public benchmarks like QPLIB and Netlib. While I understand that this presents significant challenges for current methods, a short discussion on the proposed model’s ability or limitations in handling these cases would be interesting.
> 2. Regarding the fairness of comparison, I don't believe it's a sufficient justification to omit a comparison with the GPU-based PDLP. Integrating their Julia implementation with the current Python setup would require only minor modifications to the code. link: https://github.com/jinwen-yang/cuPDLP.jl

---

> > ### Author Response · Authors · 2025-08-06
> >
> > Thank you for your thoughtful feedback and continued engagement with our work. In response, we provide an expanded discussion on solver generalization and fairness of comparisons, along with new empirical results that address recent developments in GPU-accelerated baselines. We hope these clarifications and additions fully address the reviewer's concerns.
> >
> > 1. We appreciate the reviewer’s thoughtful comment. On the question of numerical ranges: we do not expect this to pose a significant challenge as problem data can often be normalised to a given numerical range -- a common practice in classical solvers to improve numerical stability and convergence [arXiv:1312.3039]. We can therefore assume that such normalisation is applied prior to using our model. That said, public benchmarks such as QPLIB and Netlib often include problems that are poorly scaled, infeasible, or contain redundancies. Our current method does not explicitly address these scenarios, which indeed limits its applicability as a fully general-purpose solver. Addressing poor scaling or infeasibility remains an important direction for future work, requiring additional mechanisms such as constraint preprocessing or adaptive scaling techniques. We will clearly acknowledge these limitations in the revised manuscript.
> >
> >     We would like to clarify that the goal of this work is not to develop a universal solver. Rather, our aim is to introduce a GNN-based architecture that guarantees feasibility with theoretical backing which can serve as a foundation for a broad range of downstream applications. Optimization tasks represent just one such example.
> >
> > 2. Regarding the fairness of comparisons, we agree with the reviewer that a GPU-based PDLP baseline is important. At the time of submission, and as we noted in our earlier response, GPU acceleration for PDLP was not publicly accessible via Python. We only became aware after our initial rebuttal that CVXPY has recently added support for NVIDIA’s cuOPT package, enabling straightforward access to GPU-accelerated PDLP from Python. Using this interface, we have collected additional results, which we summarize in the table below. We compare GPU PDLP to ProjNet (with $c_h =1$) across problem sizes $n = 10^4$ to $10^6$, reporting runtime as a fraction of ProjNet’s:
> >
> > \begin{array}{c|c c c c c c c c c}
> >     \epsilon\overset{\LARGE\setminus}{\phantom{.}}\overset{\Large n}{\phantom{l}} & 1 & 2 & 3 & 4 & 5 & 6 & 7 & 8 & 9 \\\\ \hline
> >     10^{-4} & 0.51 & 0.59 &  0.46 & 0.44 & 0.44 & 0.41 & 0.50 & 0.59 & 0.42 \\\\
> >     10^{-3} & 0.52& 0.43& 0.51& 0.76& 0.87& 0.97& 1.03& 1.04& 0.99\\\\
> >     10^{-2} & 0.76& 0.78& 0.85& 1.29& 1.41& 1.53& 1.5 & 1.61& 1.54
> > \end{array}
> >
> >   These results show that GPU PDLP is very competitive, especially at low tolerances. Nevertheless, ProjNet still scales better for larger problem sizes and higher tolerances, achieving speed-ups of over 50\% in these settings. We believe this highlights the scalability and practical efficiency of our framework, particularly given that ProjNet is a general, objective-agnostic method, while PDLP is a state-of-the-art, specialized LP solver. We will incorporate these findings and their implications in the final manuscript.

---

> > > ### Comment · Reviewer_ev9W · 2025-08-07
> > >
> > > I appreciate the author’s thoughtful and thorough response. All of my concerns have been adequately addressed, and I will maintain my original score.

---

### Official Review · Reviewer_SUYk · 2025-06-24

**Clarity:** 3
**Significance:** 3
**Originality:** 2
**Rating:** 4
**Confidence:** 2

**Summary:**

The paper introduces ProjNet, a novel GNN framework for enforcing input-dependent convex constraints via a CAD algorithm and sparse vector clipping. A theoretical result on the convergence of the CAD algorithm is provided, and numerical comparisons with classical baselines are presented.

**Questions:**

No.

**Ethical Concerns:**

["NO or VERY MINOR ethics concerns only"]

**Final Justification:**

The authors addressed my concerns about related works and ML baselines. I increase the rating to 4.

**Limitations:**

The authors present limitations using numerical experiments.

**Paper Formatting Concerns:**

No.

**Quality:**

3

**Strengths And Weaknesses:**

__Strengths:__

1. This paper is in general well-written and is easy to follow. The main ideas are smooth and reasonable.

2. A theoretical result on the convergence of the CAD algorithm is provided.

3. Surrogate gradients are proposed as a computationally efficient alternative to exact gradients, with some theoretical properties.

4. The numerical results look nice compared to classical baselines.

__Weaknesses:__

1. I do not think that projecting the neural network output onto a feasible region is a completely novel idea. There have been some existing works, for example: Wang, R., Zhang, Y., Guo, Z., Chen, T., Yang, X., & Yan, J. (2023, July). LinSATNet: the positive linear satisfiability neural networks. In International Conference on Machine Learning (pp. 36605-36625). PMLR.

2. There are only numerical comparisons with classical baselines. Given the claimed scope of this paper, I would expect to see how this approach can contribute to the "learning-to-optimize" community. In particular, I think the authors should include comparisons with existing GNN methods for optimization. This might be a major flaw.

---

> ### Author Rebuttal · Authors · 2025-07-29
>
> We thank the reviewer for their thoughtful comments and for acknowledging the clarity and motivation of the paper, as well as the theoretical and empirical contributions. We also appreciate the points raised regarding novelty and comparisons to existing ML-based optimization methods. We address these concerns below and have revised the manuscript to better highlight our contributions relative to prior work.
>
> $\textbf{Weaknesses:}$
>
> $\textbf{Novelty:}$ We agree that the idea of projecting neural network outputs onto a feasible set is not novel, and we do not claim otherwise. As the reviewer correctly notes, this concept has been explored in prior work, including [56], which we cite on page 2 of the manuscript.
>
> Our contribution lies in the method used to compute these projections. Specifically, we propose a novel approach that leverages the CAD algorithm to perform projections in a scalable manner using GPU-acceleration. This contrasts with most existing methods, which rely on classical solvers that are less amenable to GPU implementations, especially for sparse problems. The effectiveness of our approach is demonstrated by the No CAD baseline in Table 1 in the manuscript, which shows that replacing CAD with a classical solver significantly degrades performance.
>
> $\textbf{ML baselines:}$ We thank the reviewer for raising this point. In addition to classical baselines, we included three ML-based baselines in the form of ablation studies described in Appendix D.5. These include variations of our method with replacements for the proposed CAD and SVC layers, helping isolate the impact of each component.
>
> Regarding GNN-based methods for constrained optimization, we carefully considered relevant work, including [3, 21, 54]. However, such approaches typically fall into two categories:
>
> 1. Methods which rely on classical solvers internally, such as differentiable convex optimization layers [1, 3, 12]. These often use interior-point solvers during training or inference. Since we benchmark directly against state-of-the-art solvers (e.g., Gurobi), we can confidently say that our method outperforms any ML approach that embeds such solvers. Moreover, unlike CAD, these methods often lack support for both sparse constraints and GPU acceleration, rendering them unsuitable for our problem setting.
>
> 2. Inexact ML methods that use penalty terms or Lagrangian-style optimization [21, 54] are typically faster but lack feasibility guarantees, and their constraint satisfaction often degrades with problem scale. To demonstrate this, we implemented a penalty-based method obtained by isolating and training only the GNN component of ProjNet, which was trained using a soft constraint loss of the form $c_h \|w - P_C(w)\|^2$, with $c_h=1$. This allows us to assess how a purely learned, penalty-driven method performs in our specific setting. Linear programming results are shown in the table below, where infeasibility measures $\max(|Ax - b|)$, and problem sizes range from $10^4$ to $10^6$, consistent with Figure 2 in the paper.. While this method is faster, it does not guarantee feasibility. Compared to ProjNet, the penalty-based GNN results in constraint violations that are several orders of magnitude larger, despite only modest gains in speed. For example, at $\epsilon = 10^{-4}$, this method is roughly 6× faster than ProjNet but up to 10,000× less accurate in feasibility. Note that optimality $>1$ as constraints are not satisfied.
>
>     $$
>     \begin{array}{c|c c c c c c c c c}
>     \text{Problem size} & 1 & 2 & 3 & 4 & 5 & 6 & 7 & 8 & 9 \newline \hline
>     \text{Infeasibility} & 0.63 & 0.65 & 0.72 & 0.7 & 0.77 & 0.81 & 0.86 & 0.97 & 1.01 \newline
>     \text{Time} (s) & 0.009 & 0.010 & 0.018 & 0.030 & 0.055 & 0.102 & 0.184 & 0.335 & 0.601\newline \hline
>     \text{Optimality} &&&&& 1.09
>     \end{array}
>     $$
>
>     To directly address the reviewer’s concern about contributing to the learning-to-optimize community: our work presents a learned projection architecture that generalizes across input-dependent constraint sets and significantly outperforms baselines in both accuracy and efficiency. We believe this contributes a novel and practical direction for integrating deep learning with constrained optimization in a principled way. Furthermore, since ProjNet can be trained using any (potentially supervised) loss function, its applicability extends beyond the learning-to-optimize domain to broader constrained machine learning tasks—for instance, the task from [50].

---

### Official Review · Reviewer_dghc · 2025-07-02

**Clarity:** 3
**Significance:** 3
**Originality:** 3
**Rating:** 4
**Confidence:** 3

**Summary:**

The authors propose a new framework for learning GNNs that satisfy complicated convex constraints. The key idea of the framework is to develop and integrate two kinds of layers: 1) a GPU-friendly iterative algorithm, the Component-Averaged Dykstra (CAD) algorithm and 2) sparse vector clipping layers, with a base GNN. The authors demonstrate convergence results for CAD and propose a numerical scheme to compute a surrogate gradient of the CAD derivative. The method is validated on a range of convex and nonconvex constrained optimization problems.

**Questions:**

How does your method compare to existing ml / GNN-based optimization frameworks?

**Ethical Concerns:**

["NO or VERY MINOR ethics concerns only"]

**Final Justification:**

I maintain my support of this paper. They provide a novel analysis of a neural CAD algorithm, which outperforms existing methods. Scalability is benefitted by carefully designed sparse subroutines. A follow up response from the authors has provided more experiments comparing to dense differentiable optimization layers.

**Limitations:**

yes

**Quality:**

3

**Strengths And Weaknesses:**

**Strengths**

The paper is well-written and the relevant works are sufficiently described. The algorithm is novel and comprehensively analyzed- including runtime and convergence properties. Numerically efficient variants of the framework are introduced and evaluated empirically. Experimentally, the framework exhibits promising results on several benchmark problems.

**Weaknesses**

The comparative experiments are lacking. Although some classical methods are compared to, no comparison is made to existing ML-based optimization methodologies, e.g. those outlined in the related work section (differentiable optimization layers or other iterative methods). The comparison is primarily based on ablative experiments for the various components of ProjNet. I think it would be interesting and important to demonstrate ProjNet’s performance in comparison to these kinds of algorithms.

---

> ### Author Rebuttal · Authors · 2025-07-29
>
> We thank the reviewer for their thoughtful and constructive comments. We are pleased that you found the paper well-written and appreciated the novelty, analysis, and empirical evaluation of our method. Regarding comparisons to existing ML-based optimization frameworks, we appreciate the suggestion and clarify our rationale and design choices below. We have also revised the manuscript to better emphasize these distinctions.
>
> $\textbf{Weaknesses:} $
>
> We appreciate the reviewer’s suggestion to include comparisons to ML-based optimization methods. In addition to the three ML baselines already included in the paper, we actively explored a number of other potential methods.
>
> $\textbf{Differentiable optimization layers:}$ We experimented with the qpth solver [3], a differentiable QP layer used in several prior works. However, it does not support sparse constraints as CAD does, and relies on dense matrix operations. As a result, it was not possible to scale to the problem sizes we consider. Due to these scaling issues, it was significantly slower, and for larger instances, could not fit in memory. Given these limitations, we found it unsuitable as a practical baseline, particularly since ProjNet succeeds in both speed and scalability.
>
> $\textbf{Penalty-based ML methods:}$ motivated by penalty-based ML methods, such as [21], we also tested a penalty-based variant by isolating and training only the GNN component of ProjNet, which was trained using a soft constraint loss of the form $c_h ||w - P_C(w)||^2$.  This allows us to assess how a purely learned, penalty-driven method performs in our specific setting. Linear programming results are shown in the table below where infeasibility measures $\max(|Ax - b|)$, and problem sizes range from $10^4$ to $10^6$, consistent with Figure 2 in the manuscript. While this method is faster, it does not guarantee feasibility. Moreover, infeasibility errors grow rapidly with problem size, and solutions violate constraints by orders of magnitude more than ProjNet. For example, at $\epsilon = 10^{-4}$, this method is roughly 6× faster than ProjNet but up to 10,000× less accurate in feasibility.
>
> $$
> \begin{array}{c|c c c c c c c c c}
>     \text{Problem size} & 1 & 2 & 3 & 4 & 5 & 6 & 7 & 8 & 9 \newline \hline
>     \text{Infeasibility} & 0.63 & 0.65 & 0.72 & 0.7 & 0.77 & 0.81 & 0.86 & 0.97 & 1.01 \newline
>     \text{Time} (s) & 0.009 & 0.010 & 0.018 & 0.030 & 0.055 & 0.102 & 0.184 & 0.335 & 0.601\newline \hline
>     \text{Optimality} &&&&& 1.09
> \end{array}
> $$
>
> $\textbf{Other iterative methods:}$ We also considered simple iterative algorithms for handling constraints, such as Sinkhorn's algorithm [19, 50] and related methods [56]. However, these are typically designed for specific constraint types. Sinkhorn’s algorithm applies primarily to doubly stochastic matrix constraints and is not applicable to our setting, which involves sparse, input-dependent constraints.
>
> $\textbf{Questions:}$
>
> Existing GNN- or ML-based optimization frameworks typically fall into two categories:
>
> 1. Methods that use classical solvers internally, either during training (e.g., through differentiable optimization layers) or inference. Examples include convex optimization layers [1, 3, 12], which explicitly rely on classical solvers such as interior-point methods. Since we directly benchmark against a state-of-the-art classical solver (i.e., Gurobi), our results apply to any ML method that incorporates such solvers in its pipeline.
>
> 2. Inexact or penalty-based ML methods which attempt to enforce constraints through soft penalties. These methods do not guarantee feasibility and often yield solutions that violate constraints, particularly at scale. As discussed in the weakness section above, we tested such an approach using the GNN component of ProjNet trained with a penalty loss. This method was faster but significantly less accurate, with constraint violations several orders of magnitude larger than our method (see table above).
> In contrast, our approach ensures hard constraint satisfaction while maintaining competitive runtime, and does so without reliance on classical solvers during inference. To our knowledge, no existing ML-based method matches this combination of feasibility, scalability, and efficiency in our setting.

---

> > ### Comment · Reviewer_dghc · 2025-08-06
> >
> > I thank the authors for their comprehensive and detailed response to my review and I appreciate the addition of a new experiment. Although I believe a comparison to a differentiable optimization layer would strengthen the paper, I understand that it may not be feasible to run such an experiment on the tight author-reviewer discussion period. I appreciate the authors efforts and their new experiment on the penalty-based method for generating feasible outputs. I do think there is value in this paper so I maintain my support for its acceptance.

---

> > > ### Author Response · Authors · 2025-08-07
> > >
> > > We thank the reviewer for their thoughtful follow-up. We agree that including a differentiable optimization layer baseline would enhance the completeness of the comparisons, and we did make efforts to implement one during the development of this work. However, we are currently only aware of two relevant methods. The first is qpth [3], which -- as noted -- does not support sparse problems, making it impractical for the large-scale settings considered in our paper. The second is the DPP method [Differentiable Convex Optimization Layers, arXiv:1910.12430], which implements a differentiable conic programming layer built on top of a CPU-based solver and assumes fixed input dimensions (i.e., batched problems with fixed $n$ and $m$). These assumptions are incompatible with GNNs, which operate over graphs with varying structure and size.
> > >
> > > We did implement qpth, but due to its dense formulation, it exhibited extremely poor performance (orders of magnitude slower than other baselines) which would not have constituted a meaningful or fair comparison. While the DPP method does support sparse problems, its reliance on fixed input shapes prevents batching on graph data, making training infeasible. Furthermore,  DPP embeds classical solvers similar to those already included in our benchmarks and is therefore strictly slower than them.
> > >
> > > We believe this highlights a practical gap: current differentiable solvers are not well-suited to the sparse, heterogeneous problem structures that arise in graph-based settings. We will further highlight these observations in the final version to better motivate our approach.

---

### Note · Authors · 2025-08-12

We thank the reviewers for their thoughtful engagement with our paper. The discussion provided valuable feedback that helped strengthen the work. The main concern raised was the limited inclusion of ML baselines, as our comparisons focused on classical optimization methods. We hope we clarified the rationale: there is a scarcity of ML methods that offer hard feasibility guarantees for the heterogeneous, instance-dependent constraints we considered. Accordingly, we evaluated ProjNet on traditional optimization problems where it was possible to benchmark against specialized solvers. While it is unusual to apply an end-to-end method like ProjNet to such problems, it allowed us to validate our methods against state-of-the-art solvers. Had we chosen more complex tasks, the results would likely have been even more polarizing due to the absence of directly comparable methods.

Since submission, we have added results for the GPU implementation of PDLP as an additional baseline for linear programming. Our new experiments show that while PDLP is competitive at low tolerances, ProjNet remains faster for larger problems and higher tolerance (sometimes by over 50\%) despite being a general, objective-agnostic method. This further demonstrates the scalability and efficiency of our framework, even against specialized solvers.

We also wish to highlight a contribution underplayed in our submission: to our knowledge, we provide the first convergence result for CAD. In [14], it was claimed —without proof — that CAD’s component-averaging mechanism would simply accelerate Dykstra’s algorithm without changing its limit point. We showed that this does not hold in general and supplied the missing convergence theory. In addition, we presented a batched, GPU-accelerated CAD implementation that is competitive with, and in some regimes outperforms, specialized solvers.

We believe these theoretical and algorithmic contributions stand on their own merit, independent of the ML component of the work. Combined with the new PDLP comparisons, they show that ProjNet is not only conceptually novel but also practically competitive, offering a good blend of rigorous theoretical guarantees and strong empirical performance.

---

### Decision · Program_Chairs · 2025-09-17

**Decision:**

Accept (poster)

**Comment:**

This paper introduces ProjNet, a GNN architecture designed to enforce input-dependent constraints. To implement this architecture, Sparse Vector Clipping and Component-Averaged Dykstra are proposed for refined solution qualities. The paper provides a theoretical convergence guarantee for CAD and implements it with GPU acceleration. The main concern is the bottleneck of CAD component. In summary, all the reviewers agree that borderline accept would be reasonable and fair.